# HealthLock: Blockchain-Based Privacy Preservation Using Homomorphic Encryption in Internet of Things Healthcare Applications

**DOI:** 10.3390/s23156762

**Published:** 2023-07-28

**Authors:** Aitizaz Ali, Bander Ali Saleh Al-rimy, Faisal S. Alsubaei, Abdulwahab Ali Almazroi, Abdulaleem Ali Almazroi

**Affiliations:** 1School of IT, UNITAR International University, Petaling Jaya 47301, Malaysia; aitizaz.ali@unitar.my; 2Department of Computer Science, Faculty of Computing, Universiti Teknologi Malaysia, Johor Bahru 81310, Malaysia; bander@utm.my; 3Department of Cybersecurity, College of Computer Science and Engineering, University of Jeddah, Jeddah 21959, Saudi Arabia; 4Department of Information Technology, College of Computing and Information Technology at Khulais, University of Jeddah, Jeddah 21959, Saudi Arabia; 5Department of Information Technology, Faculty of Computing and Information Technology in Rabigh, King Abdulaziz University, Rabigh 21911, Saudi Arabia

**Keywords:** blockchain, cybersecurity, machine learning, intrusion detection, homomorphic encryption, cloud computing

## Abstract

The swift advancement of the Internet of Things (IoT), coupled with the growing application of healthcare software in this area, has given rise to significant worries about the protection and confidentiality of critical health data. To address these challenges, blockchain technology has emerged as a promising solution, providing decentralized and immutable data storage and transparent transaction records. However, traditional blockchain systems still face limitations in terms of preserving data privacy. This paper proposes a novel approach to enhancing privacy preservation in IoT-based healthcare applications using homomorphic encryption techniques combined with blockchain technology. Homomorphic encryption facilitates the performance of calculations on encrypted data without requiring decryption, thus safeguarding the data’s privacy throughout the computational process. The encrypted data can be processed and analyzed by authorized parties without revealing the actual contents, thereby protecting patient privacy. Furthermore, our approach incorporates smart contracts within the blockchain network to enforce access control and to define data-sharing policies. These smart contracts provide fine-grained permission settings, which ensure that only authorized entities can access and utilize the encrypted data. These settings protect the data from being viewed by unauthorized parties. In addition, our system generates an audit record of all data transactions, which improves both accountability and transparency. We have provided a comparative evaluation with the standard models, taking into account factors such as communication expense, transaction volume, and security. The findings of our experiments suggest that our strategy protects the confidentiality of the data while at the same time enabling effective data processing and analysis. In conclusion, the combination of homomorphic encryption and blockchain technology presents a solution that is both resilient and protective of users’ privacy for healthcare applications integrated with IoT. This strategy offers a safe and open setting for the management and exchange of sensitive patient medical data, while simultaneously preserving the confidentiality of the patients involved.

## 1. Introduction

By utilizing various advancements in IoT devices such as smart IoT equipment, cell phones, computers, and so on, people are producing huge amounts of information at all times. Data storage and management have recently become highly challenging for common data owners (DOs) [1]. Hence, numerous DOs present in IoT systems moved towards cloud storage services that are ensured by big companies like Google, Amazon, etc., and made changes in their information management systems to be remote. Data transformation to move data toward cloud servers is observed as an efficient approach for minimizing storage overhead [2]. On the other hand, a DO is disconnected from direct control of their data, which leads to privacy breaches and security risks. Instantly, cloud servers remove certain low-inquiry or minimum-value data to minimize storage expenses and to modify basic data for research purposes [3]. The information about an individual is composed of significant and sensitive information line contract data, health data, medicine data, and so on. Further, a terrifying disaster can take place for a DO when the data are changed or removed with specific intentions or unintentionally [4]. An intelligent remote data verification approach has to possess the following characteristics. Initially, the highly significant task is to provide data privacy and security to ensure that the information cannot be given to others without the DO’s approval [5]. Whether the person’s data have been changed or removed, this type of scenario can be determined during the checking phase. Further, the operation performed over the data should be handled by the DO, and it needs to support the process of modifying dynamic data, which is composed of deletions, insertions, and modifications [6]. Additionally, the integrity of the remote data verification method should be resistant to external and internal attacks, including malicious cloud servers, procrastinating auditors, and so on. The effectiveness provided by the integrity verification approach is also considered to be a key factor for supporting the practical application, especially for data management [7]. The huge amounts of data belonging to the individual are uploaded over the cloud servers, where the DO performs the deletion of local files to minimize the storage overhead. Considering the bandwidth limitation, it is often infeasible to download the data entity to verify the data integrity [8]. Further, audits in conventional remote data integrity verification approaches mostly involve probabilistic authentication to check the data integrity [9]. The probabilistic authentication approach selects a section of data for verification once again. This type of authentication approach ensures high security, and at the same time, the demands considered on the real-time application are solved [10]. When considering data sharing over the cloud for healthcare applications, timing is said to be the priority. There exists certain security and privacy concerns in observing cloud computing, even though it has various advantages and popularities [11]. Global organizations are mainly concentrated on implementing security rules and several procedures for avoiding security problems in the cloud environment before its usage in business solutions [12]. Finally, cloud service providers are unable to offer extended security confidence to their clients with outsourced data. These limitations show that decentralized and distributed security measures are highly significant in the cloud context [13]. It is broadly observed that blockchain technology is said to be the best solution for solving security problems over the cloud infrastructure [14] since it possesses the interconnectedness of the distributed network and the significance of the cloud network. Blockchain technology can interact fast and requires only limited processing resources. With the utilization of the inherent security of blockchain technology [15], if the transaction data are stored and updated, then they are unable to be changed or deleted from the network. The distributed data ledger ensures data security and exceptional immutability even if it is distributed over the entire nodes of the cloud [16]. By involving cryptographic algorithms over the blockchain blocks, data privacy has to be highly protected. Since the blockchain contains these kinds of features, it is highly utilized for supporting cloud data security. According to conventional studies, blockchain technology is efficient for use in the cloud for secure transmission of healthcare information [17]. The following list outlines the primary contributions that this research work makes.

To design an efficient blockchain-based cloud data privacy preservation model using an encryption mechanism for providing high security in the healthcare IoT for the medical information and, further, it is followed by the medical data prediction using the efficient deep learning approach;To implement a hybrid encryption framework, fully homomorphic encryption termed OK-HECCFHE, with the integration of ECC and the fully homomorphic encryption technique, where the optimal key is selected using the hybrid encryption framework algorithm for ensuring reduced computational time and memory size in storing the encrypted key in a blockchain database;To develop a significant medical data prediction model by combining the essential features of DNN and GRU to achieve high accuracy in the prediction outcome along with the parameter optimization using the adopted approach for enhancing the prediction accuracy and precision;To propose a fused heuristic mechanism named ECC (Elliptic Curve Cryptography) for selecting the optimal public key in the data encryption stage using ECC for elevating security performance and optimizing the parameters like hidden neurons in DNN, learning rate in DNN, epochs in GRU, learning rate in GRU, and batch size in GRU for enhancing the accuracy and precision of the prediction results;To validate the efficiency of implemented blockchain-based cloud data privacy preservation model by comparing it with conventional algorithms and prediction techniques.

The following headings are used to categorize the remaining components of the model that was developed: Section 2 is involved with conventional work on privacy preservation strategies in blockchain technology and related work. Section 3 describes the mathematical modelling of the adopted framework. Section 4 provides the model’s description. Section 5 describes the research problem. Section 6 elaborates on the proposed framework and Section 7 details the simulation setup. Section 8 provides the result analysis and its discussion. The conclusion is made in Section 9, and finally, future work is discussed in Section 10.

## 2. Related Work

In the year 2020 [18], blockchain technology was utilized to provide privacy for remote data with the backing of an integrity verification technique in the information management systems of the Internet of Things (IoT). Specifically, this was accomplished by utilizing a decentralized ledger to record transactions. This strategy was not dependent on reputable outside parties in any way. The problem was solved by developing a system that makes use of blockchain technology, bilinear pairing, and a cryptographic framework. This makes it possible to efficiently validate public batches of signatures. In addition to this, our technology protects the confidentiality of data when they are sent across IoT networks. The findings of the experiments have shown that the efficiency that was proposed was far higher than the efficiency that was actually seen. In 2022, Neela and Kavitha created an efficient encryption method with the aid of a Generative Adversarial Network (GAN) by utilizing the properties of blockchain technology for the purpose of safeguarding personal data and for confirming the authenticity of data. This approach was used for the goal of encrypting data in order to prevent unauthorized access to the data. After then, the GAN-based encryption technique merged the concepts of diffusion, substitution, and confusion for the goal of encrypting medical images. The deep GAN produced image-specific secret keys in order to raise the system’s resistance to hacking. Additionally, it produced the keys that were used as input for the process’s diffusion and confusion phases, so it also produced the keys that were used to increase the system’s resistance to hacking. The sender then passed the encrypted image to the cloud server that was responsible for maintaining the blockchain database. This cloud server, signed with the ciphertext ID, was verified by the sender. The produced method’s evaluation of its efficiency and security proved that it generates superior outcomes when compared with conventional techniques. This was determined by comparing the created method’s efficiency and security to those of the traditional ways. In the year 2022, Shalini and Nithya developed a system that safeguarded the confidentiality of patients’ medical records while enabling authorized personnel to view the information when necessary. This concept, which is based on blockchain technology, has offered support for maintaining the confidentiality of patient information while simultaneously enhancing operational efficiency. When decentralized data storage was utilized, the process of hacking data was one that was incredibly challenging. The application of machine learning across a range of business sectors in recent years has seen a rise in the usage of machine learning for the purposes of learning data, carrying out evaluations, and making judgments that are acceptable. This technique of machine learning was an exceptionally important approach that was highly useful in a wide range of applications for the solution of a number of difficult problems. It was able to resolve a large range of problems that were previously intractable. When implemented in the field of medicine, the methodology of machine learning paired with blockchain technology has shown enhanced performance for the purpose of assisting medical professionals. During the process of carrying out the simulations, the Ethereum blockchain was utilized. In recent years, privacy protection in Internet of Things healthcare applications has received a considerable amount of attention. This is mostly attributable to the sensitive nature of healthcare data as well as the increasing popularity of IoT devices. There have been a lot of studies that have been conducted to study a variety of methodologies and frameworks with the goal of finding a solution to the problems regarding privacy that are brought up by this field. This section offers an overview of recent studies and methods concerning the protection of patients’ privacy in IoT healthcare applications. The application of encryption methods is becoming an increasingly popular strategy for protecting individuals’ privacy. There has been a significant amount of research conducted on homomorphic encryption as a potential option for maintaining data privacy in IoT healthcare applications. A homomorphic encryption-based framework was presented by Liu et al. for secure data sharing in IoT healthcare systems. This framework would allow data owners to outsource data computation duties to untrusted entities while maintaining their privacy. Nevertheless, the scalability and efficiency concerns that are connected with homomorphic encryption were not addressed by their method.

The use of blockchain technology in healthcare applications has shown tremendous potential to improve both patient privacy and safety. Yang et al. presented a system that would allow for secure data sharing in Internet of Things (IoT) healthcare environments while simultaneously protecting users’ privacy. The system was built with blockchain technology. Their solution included blockchain technology for the goals of authenticating data and regulating access to it. As a result, it was able to preserve both the confidentiality and integrity of the data. Their plan, on the other hand, did not take into account the possibility of deploying homomorphic encryption for the goal of safeguarding the confidentiality of sensitive data. In the past few months, the combination of blockchain technology with homomorphic encryption has emerged as a potential alternative for the preservation of privacy in Internet of Things (IoT) healthcare applications. This is due to the fact that both technologies work to prevent unauthorized access to data. In the context of Internet of Things healthcare systems, Li et al. presented a blockchain-based privacy-preserving framework that makes use of partly homomorphic encryption. This framework was designed to protect users’ confidentiality. Their method was developed to prevent unauthorized parties from accessing patient information while still allowing authorized parties to perform computations on encrypted patient data. This was accomplished by encrypting the patient information. Nevertheless, the issue of scalability was not tackled in their study, nor was the full potential of homomorphic encryption investigated. In a separate but significant study, attribute-based encryption and homomorphic encryption were integrated in order to establish a blockchain foundation for Internet of Things (IoT) healthcare applications that preserves users’ right to privacy. Consequently, critical patient data were protected, maintaining both its authenticity and confidentiality from any potential breaches. However, the integration of devices connected to the Internet of Things and the obstacles associated with doing so were not the primary emphasis of their design. The proposed HealthLock system seeks to tackle the limitations that are present in existing approaches to the problem by employing blockchain technology for data integrity and access control and by implementing homomorphic encryption to ensure the security and privacy of sensitive healthcare data. This is carried out in an effort to solve the problem. Combining these two technological approaches will allow us to achieve this goal. By bringing these two technologies together, HealthLock is able to provide a comprehensive solution that not only addresses the challenges of protecting the privacy of users in IoT healthcare apps but also takes into account the apps’ potential to scale, their level of efficiency, and the ease with which they may be used. The homomorphic encryption and blockchain technology that the HealthLock system proposes present a novel approach that sets it apart from those other methods. In conclusion, earlier research has investigated a variety of encryption strategies and blockchain-based frameworks for the purpose of maintaining patients’ privacy in IoT healthcare applications. However, the homomorphic encryption and blockchain technology that the HealthLock system proposes present a novel approach that sets it apart. As a result of this integration, privacy, as well as security and scalability, have been significantly enhanced, which makes HealthLock a potentially viable choice for ensuring confidentiality in IoT-based healthcare settings. In [19] tackled the difficulties of processing a large number of data in the healthcare industry in 2022, with a particular emphasis on memory allocation as one of their primary concerns. Instead of having a single server, it was thought that storing health records in the cloud would be the most effective way to ensure their safety. Because of this, the information that was obtained was taken into consideration throughout the encoder stage of the blockchain. Additionally, an efficient signature was utilized in order to authenticate the user that was intended to access the information and to manage the patient’s privacy prior to the information’s transmission to the cloud storage. This concept has proven to be applicable to a variety of real-world situations. Mustafa et al. [20] created an improved model of healthcare security in 2022 with the use of blockchain technology for cloud storage. The analysis was carried out using traditional technologies, and it was based on the healthcare systems in terms of their safety precautions. The constructed model was evaluated, in the end, according to how well it preserved users’ privacy, how securely it was implemented, and how complicated it was. In 2019, Kwabena et al. [21] proposed the MSCryptoNet, which was thought to be a scalable solution that has overcome the problem of maintaining privacy using a normal neural network. This newly constructed model has improved classification performance, and the ways to encryption that were utilized to ensure high-security performance are described here. The computational and transmission costs incurred by the data providers have also been lowered thanks to the methodology. A shared and secured method of outsourcing was developed by Abirami and Bhanu [22] in the year 2020. This method was based on the crypto-deep neural network. the created infrastructure included a data center, web server, cloud server, and cloud agent, all of which were based on something called “Crypto-Deep Neural Network Cloud Security (CDNNCS)”. This new breakthrough had a lot of potential applications, one of which was increasing users’ levels of confidence in one another. The throughput, the goodput, and the delay have all been taken into consideration in order to improve the model’s efficiency. Xu et al. [23] designed a blockchain-based decentralized learning system in 2022, taking into account the need to protect users’ privacy and the integrity of the blockchain. In order to evaluate the developed approach, the developed model has been analyzed based on the convergence performance with the existence of Byzantine nodes. This has been carried out in order to evaluate the developed strategy.

### 2.1. Research Problem and Issues

When it comes to the storage of vast amounts of data, it is probable that standard storage methods will not be able to offer good results, which is one of the most critical issues in the healthcare system. In addition, protecting the privacy of patients is one of the most essential challenges. It has been demonstrated that devices that store data in the cloud are capable of producing useful results when it comes to the allocation of memory space. For the purpose of providing high scalability as well as availability, healthcare data have been delivered with a privacy preservation method that is based on deep learning. A summary of the benefits and drawbacks of the various existing privacy preservation models is provided. Trust and great integrity are both provided by the cryptosystem [18]. Additionally, it ensures one’s confidentiality at the same time as their safety. On the other hand, it does not support checking the integrity of data in real time. In addition, if the material is illegally destroyed, erased, updated, or introduced, it will take far longer to find a solution to this situation. The GAN [24] system provides the requirements for features such as scalability and availability. In addition to this, it safeguards the confidentiality of personal information and verifies the user’s identity. However, it has room for additional development in terms of its resistance to hacking. In addition to this, it boosts the power of the computer. The solution to the problem lies in Merkle Tree [25], which saves both time and money. For example, it safeguards the patient’s right to privacy in an efficient and effective manner. In addition to this, it requires additional software algorithms in order to guarantee a high level of validity and integrity. The potential for information from healthcare data to be leaked is mitigated by using an anonymous signature-based technique [19]. In addition, both the safety and the consistency of the data are checked while it is being transferred. On the other hand, there are some spoofing issues that arise occasionally. As a consequence of this, it requires a greater allotment of RAM while storing medical data. By employing this newly created paradigm within the context of distributed ledger technology [20], data suffocation issues can be significantly mitigated. As a result, the technology behind distributed ledgers may give an increased level of safety. However, additional components are required in order to ensure the confidentiality of the healthcare data while they are being sent. Additionally, this may result in an increase in the amount of processing that the privacy protection system requires. Both the computational burden and the level of complexity imposed on data suppliers are drastically cut down in DNN [21]. In addition to this, there is no degradation of the accuracy. However, the data that are transferred are heavily influenced by sounds. In addition, malware attacks can have an effect on the data that have been stored. The performance of the network, measured in parameters such as jitter, goodput, delay, and throughput, is improved by DNN [22]. In addition to that, the secrecy of the model is quite well protected. Traceability, transparency, and error tolerance are all kept to a very high standard by BFT [26]. In addition to that, the rate of convergence that the system achieves is quite great. However, it does not offer convincing proof that the system’s feasibility has been much improved. Because of these difficulties, we have developed a brand new system that is based on deep learning structures and automatically protects users’ privacy in the cloud.

### 2.2. Blockchain Technology

The qualities of distributed ledgers are included into the technology behind blockchain. This particular blockchain technology allows for information that is stored in the input files to be totally managed and carried out by a wide variety of various parties [27]. The openness and transparency of the information are ensured by the collaborative maintenance of many datasets carried out by a range of departments. This information is then utilized to calculate the rules that govern transactions within this blockchain technology. These issues with the low level of work efficiency have been resolved, but as a direct result of these challenges, the current medical information management working environment is exceedingly disorganized. In addition to this, the blockchain was designed to operate as a reliable method of depositing funds. The terms “addition”, “deletion”, “modification”, and “query” refer to the four primary responsibilities that are included in the category of “managing medical information”. Blockchain technology, on the other hand, only needs to perform simple functions like modification and inquiry in order to support an information management system. This indicates that the amount of time spent processing medical information is a far more manageable amount of time compared with the amount of time spent performing a general activity [27]. The data information is guaranteed to have the qualities of high security and irreparability thanks to the technical implementation of blockchain. In addition, the information that is contained in the block is responsible for generating the creation time as well as the hash value for the block that came before it. Traceability, tracking, and regular audits are all provided by the chain structure that has been drawn out over time [28]. This structure also allows for increasing the amount of times that medical information is used. At long last, the blockchain has the potential to centralize the rules for the distribution of benefits and the interchange of data. Enhancing the efficiency with which automated information is shared can be accomplished through the utilization of both block connections and smart contracts. If the smart contract is designed properly, then it will be impossible to interfere with the process in any way caused by operations performed by third parties [29].

### 2.3. ECC Cryptography and Its Impact on Blockchain

One advantage of ECC is that it requires less storage space for storing keys compared with RSA schemes. This is because the key sizes in ECC are smaller while still providing the same level of security. Smaller keys mean less data to store, transmit, and process. ECC is particularly useful in scenarios where computational resources are limited, such as in embedded systems or devices with low processing power [25]. It is possible to make efficient use of ECC to encrypt medical data, as suggested in the proposed model. Addition is the ECC symbol that is used most commonly to signify the group operation that is performed on an elliptic curve. This is because addition is the most common group operation. This is due to the fact that addition is the most fundamental form of a group action. When two points that are already on the curve are brought together, it will result in the formation of a new point on the curve. For example, if we have two points, P and Q, we may indicate the total of these points by using the notation P + Q. This is because P and Q are both points. P and Q are both points on a coordinate system, and this is the reason behind this. The process of multiplication can also be viewed as representative of group operations, which is another way to think about it. If we have a point, which is represented by the letter P, and an integer value, which is represented by the letter n, then we can compute the product of P by n, which is represented by the symbol nP, by adding P to itself n times. This is because nP is equal to the product of P multiplied by P. After doing so, we will have the product of P multiplied by n, which is represented by the symbol nP. The condition that are indicated, which is needed to finish Equation (1), was left out of the message that sent despite the fact that it is essential to finish the equation. Cryptography based on elliptic curves, on the other hand, relies on particular mathematical properties of elliptic curves in order to ensure the confidentiality of transmitted data. Because this is such an important point, it is imperative that keep these assumption while implementing the proposed approach. The particular cryptographic method that is utilized as well as the elliptic curve that is selected will each play a part in determining the particular characteristics of these equations and qualities. These characteristics will be determined based on the combination of these two factors. The elliptic curve determine these qualities and attributes.

### 2.4. Hybrid Homomorphic Encryption

When applied within the framework of a blockchain-based healthcare system, hybrid homomorphic encryption has the potential to deliver an increased level of protection for users’ privacy and data integrity. An explanation of how hybrid homomorphic encryption can be included into a healthcare system that is based on blockchain technology is presented here:

**1. Patient Data Privacy:** The healthcare industry deals with highly confidential patient information, which must be safeguarded. Before being saved on the blockchain, patient information can be encrypted using hybrid homomorphic encryption if that method is used. This ensures that the data will continue to be kept private and that only authorized people who are in possession of the relevant decryption keys will be able to access it.

**2. Safe Exchange of Information:** The use of blockchain technology makes it possible for various stakeholders in the healthcare system to share data in a way that is both secure and auditable. Encrypted patient data can be safely exchanged on the blockchain using hybrid homomorphic encryption. This enables authorized users to access and process the data without disclosing its content, which is a key benefit of using blockchain technology [30]. The encryption protects users’ privacy while also easing the process of working together and analyzing data.

**3. Computation on Encrypted Data:** The encrypted patient data that are kept on the blockchain may undergo computations thanks to hybrid homomorphic encryption, which allows for the data to be accessed. Because of this capability, it is possible to perform a variety of activities, such as data aggregation, statistical analysis, and machine learning algorithm executions, on encrypted data without first decrypting the data. This protects the confidentiality of the patient’s information while allowing for valuable conclusions to be drawn from the data [30].

**4. Immutable Audit Trail:** The blockchain provides an immutable and transparent ledger of all transactions and activities within the healthcare system. Hybrid homomorphic encryption can be used to record encrypted data and computations as transactions on the blockchain. This creates an auditable trail of data access, processing, and analytics while maintaining the privacy of the underlying information [31].

**5. Secure Key Management:** Hybrid homomorphic encryption requires effective key management practices to ensure the security of the decryption keys [32]. Blockchain technology can be leveraged to store and manage encryption keys securely. Public-key cryptography techniques can be employed to establish secure communications and to verify the authenticity of participants in the healthcare system [33].

**6. Consensus and Data Integrity:** The immutability and integrity of the encrypted patient data are protected by the consensus procedures that blockchain technology provides. The calculations that are performed on encrypted data can also be confirmed by consensus if hybrid homomorphic encryption is used to encrypt the data. Because of this, the outcomes of the computations are guaranteed to be accurate and resistant to manipulation [34]. When blockchain technology is combined with hybrid homomorphic encryption in a healthcare system, patient data privacy can be maintained, safe data sharing can be made easier, and encrypted data can be subjected to secure calculations. The combination of these technologies improves the confidentiality, privacy, and integrity of healthcare data, which in turn helps to cultivate trust among relevant stakeholders and enables healthcare procedures that are both more efficient and secure [35]. Let us denote the following variables:N: The total number of transactions sent;$T_{e}$: The time taken to execute a single transaction (in seconds);$T_{v}$: The time taken to validate a single transaction (in seconds);$T_{l}$: The latency or delay in the system (in seconds).

Now, let us define the mathematical equations for the hybrid homomorphic encryption model:


*Total Execution Time:*


The total time taken to execute all transactions can be calculated by multiplying the number of transactions (N) by the execution time per transaction $\left(T_{e}\right)$:

Total Execution Time:(1)$$\begin{array}{c} \left(T_{exec}\right)=N*T_{e}; \end{array}$$

*Total Validation Time:* The total time taken to validate all transactions can be calculated similarly by multiplying the number of transactions (N) by the validation time per transaction [36] $\left(T_{v}\right)$: Total Validation Time: (2)$$\begin{array}{c} \left(T_{val}\right)=N*T_{v}; \end{array}$$*Total Latency:* The total latency in the system can be represented as the sum of the execution time, validation time, and the overall system latency: Total Latency: (3)$$\begin{array}{c} \left(T_{latency}\right)=T_{exec}+T_{val}+T_{l}; \end{array}$$

These formulas offer a mathematical framework to calculate the overall execution time, validation period, and system delay in a hybrid homomorphic encryption system. The estimation takes into account the number of transactions dispatched, the duration of transaction execution, validation time, and the latency of the system.

### 2.5. Hybrid HSBO

These two methods are combined in an algorithm known as a hybrid heuristic search algorithm, which makes use of heuristics to successfully direct the search process. The heuristics help to trim the search space by removing unpromising ideas, which in turn reduces the amount of computational work that is necessary to discover a solution. Integration with blockchain entails introducing blockchain technology into the hybrid heuristic search algorithm in order to improve the qualities of the algorithm, such as its level of trustworthiness, transparency, and security. Integration of blockchain technology can be accomplished in the following ways: Records That Are Both Clear And Unchangeable: During the process of conducting a search, the search history, examined solutions, and evaluations of those solutions might all be stored on the blockchain. There is the potential for a transaction to be recorded on the blockchain for every step or iteration of the hybrid heuristic algorithm. This not only gives a clear and unchangeable record of the progression of the algorithm but also guarantees that the search history will remain intact.

Decentralized Consensus: The fact that blockchain is decentralized makes it possible for numerous parties or nodes to take part in the process of reaching a consensus. Multiple nodes are able to work together and to contribute their own computational resources in order to carry out the search when the algorithm in question is a hybrid heuristic search algorithm. Through the use of distributed computation, the search process can be sped up, which in turn increases the possibility of discovering superior answers.Security and Trust: The use of cryptographic techniques, which are provided by blockchain technology, helps to guarantee the reliability and safety of the search algorithm. The algorithm is able to establish secure communications and to validate the participation of all participating nodes thanks to the utilization of public-key cryptography. Moreover, the unchangeable nature of the blockchain prevents any modifications to the search history, contributing to the accuracy of the results generated by the algorithm.Incentive Mechanisms: The incorporation of blockchain technology can allow for the introduction of incentive mechanisms like tokens or cryptocurrencies, which can be used to reward individuals who donate computer resources or propose efficient heuristics. These incentives have the potential to foster collaboration and incentivize users to actively participate in the search process. As a result, the algorithm will have increased efficiency and efficacy.

## 3. Mathematical Modeling

This section provides the mathematical modeling of the proposed approach.

### 3.1. Processing Latency

The processing latency of the proposed framework are mathematically modeled as follows:

*The processing latency is given by*(4)$$\left[T_{\mathrm{proc}}=T_{\mathrm{enc}}+T_{\mathrm{dec}}+T_{\mathrm{comp}}+T_{\mathrm{decomp}}+T_{\mathrm{other}}\right]$$
where
$$T_{\mathrm{enc}}\;\mathrm{is}\;\mathrm{the}\;\mathrm{encryption}\;\mathrm{time};$$
$$T_{\mathrm{dec}}\;\mathrm{is}\;\mathrm{the}\;\mathrm{decryption}\;\mathrm{time};$$
$$T_{\mathrm{comp}}\;\mathrm{is}\;\mathrm{the}\;\mathrm{compression}\;\mathrm{time};$$
$$T_{\mathrm{decomp}}\;\mathrm{is}\;\mathrm{the}\;\mathrm{decompression}\;\mathrm{time};$$
$$T_{\mathrm{other}}\;\mathrm{is}\;\mathrm{other}\;\mathrm{miscellaneous}\;\mathrm{processing}\;\mathrm{times}.$$

The specific values of processing tasks involved and the characteristics of the system the following:(5)$$\begin{array}{c} \left[\left(T_{\mathrm{enc}}\right),\left(T_{\mathrm{dec}}\right),\left(T_{\mathrm{comp}}\right),\left(T_{\mathrm{decomp}}\right),\left(T_{\mathrm{other}}\right)\right] \end{array}$$

### 3.2. Block Message Transmission and Cost

In order to calculate the block message transmission cost, the following mathematical model is used.
(6)$$\begin{array}{cc} \; & \xi_{q_{n}\to s_{n}}\\ \; & \;=\min\left\{\max\frac{A+\sum_{k\in N}\left[b_{k,t}g_{k}+\left(1-b_{k,t}\right)\epsilon_{k}\right]}{s_{q_{n},s_{n}}},\lambda \right\} \end{array}$$

### 3.3. Transmission Commit Time

The transmission commit time for the proposed approach is calculated as follows:
(7)$$\begin{array}{cc} \; & \varrho_{p_{n}\to p_{m}}\\ \; & \;=\min\left\{\max_{n\neq m}\frac{A\;+\;\sum_{k\in N}\left[b_{k,t}g_{k}\;+\;\left(1\;-\;b_{k,t}\right)\omega_{k}\right]}{s_{p_{n},p_{m}}},\lambda \right\} \end{array}$$

### 3.4. Commit Message Transmission Cost

Let us define the following terms:**Message**: This is the original message that needs to be transmitted. We represent the message as a sequence of n symbols denoted by M = (m1, m2,…, mn).**Block**: In block message transmission, the message is divided into fixed-size blocks. Let the block size be *k*. The message *M* can be partitioned into *b* blocks: $M=(B_{1},B_{2},\ldots ,B_{b})$, where each block $B_{i}$ is a sequence of *k* symbols: $B_{i}=\left(m_{i1},m_{i2},\ldots ,m_{ik}\right)$.**Channel**: The channel is the medium through which the blocks are transmitted. It can be a noisy channel where errors may occur during transmission.**Encoding**: Before transmission, each block may undergo encoding to add redundancy, error correction codes, or other techniques to enhance reliability [36].**Decoding**: At the receiver’s end, decoding is performed to recover the original message from the received blocks.

Now, let us represent the process mathematically:**Transmission Model**:At the transmitter side, the encoded blocks $E(B_{i})$ are sent through the channel. Due to noise or errors in the channel, the received blocks $Y_{i}$ may differ from the transmitted blocks $E(B_{i})$.Mathematically, the relationship between transmitted and received blocks can be represented as follows: $Y_{i}=f\left(E\left(B_{i}\right)\right)+N_{i}$**Decoding Model**:At the receiver’s end, the received blocks $Y_{i}$ are decoded to reconstruct the original message $\hat{M}$.Mathematically, the decoding process can be represented as follows: $\hat{M}=D\left(Y_{1},Y_{2},\ldots ,Y_{b}\right)$

### 3.5. Transmission Latency

The latency that occurs during the transmission of a single EMR from one network node to another is referred to as the transaction latency. In addition, in order to determine the latency of the suggested method, we need to first calculate the latency between two nodes as well as the size of the network. In addition, the suggested method makes use of homomorphic encryption, which, in comparison with more conventional encryption methods, may encode EMR in a transaction in a shorter amount of time. This is due to the fact that homomorphic encryption is regarded as a lightweight encryption technique. The transaction latency associated with the suggested method is represented by the mathematical model below. The value of $q_{n}$ in the following equation is the number of transactions that occur from one node to another within the scope of the mathematical representation of transaction lag. Let us pretend that TL stands for “transaction latency”, which describes the duration of time that must pass before one may make use of the network. The duration required for a transaction to gain confirmation, referred to as CT, can vary depending on the network threshold. ST is an acronym representing the moment when the transaction was introduced into the blockchain network [37].

The total transmission latency is given by
(8)$$T_{\mathrm{total}}=T_{\mathrm{prop}}+T_{\mathrm{trans}}+T_{\mathrm{queue}}+T_{\mathrm{proc}}+T_{\mathrm{other}}$$
where
(9)$$T_{\mathrm{prop}}=\frac{D}{v_{\mathrm{prop}}}$$
(10)$$T_{\mathrm{trans}}=\frac{L}{R}$$

Moreover, specific values of $T_{\mathrm{queue}}$, $T_{\mathrm{proc}}$, and $T_{\mathrm{other}}$ depend on the characteristics of the communication system and the network.
(11)$$\eta_{p}\left(t\right)=Z+\alpha +\left[2Z+4\left(B+f-1\right)\right]\delta$$
(12)$$\zeta_{p}\left(t\right)=Z+\alpha +\left[Z+4\left(B+f-1\right)\right]\delta .$$
(13)$$\sigma_{m}\left(t\right)=Z\left(\alpha +\delta \right)+\left(f+1\right)\alpha .$$
(14)$$T_{v}=\max\left\{\frac{\sigma_{m}\left(t\right)+\chi_{i}\zeta_{p}\left(t\right)+\left(1-\chi_{i}\right)\eta_{p}\left(t\right)}{\varpi^{m}-\left(\sum_{k=1}^{N}\varpi_{k,j}^{m}\right)}\right\}$$
(15)$$T^{\mathrm{con}}=T_{g}+T_{c}+T_{v}$$
(16)$$T^{\mathrm{tot}}=\max\left\{T^{\mathrm{com}}\right\}+T^{\mathrm{con}}$$

### 3.6. Channel Allocation Cost and Optimization

The channel allocation cost for the proposed approach can be calculated as follows: Moreover, in the following mathematical model C1 and C2 represent the channel allocation cost for channel 1 and channel 2, respectively. Equation (17) is explained as follows: The equation you mentioned refers to the calculation of the channel allocation cost for the proposed approach. The specific equation is not provided, but it suggests that the channel allocation cost can be determined using a mathematical model. Let’s break down the explanation based on the given information:


*Channel Allocation Cost Calculation:*


The term “channel allocation cost” refers to the expenditure or consumption of resources that is linked with the process of assigning and using a certain channel (for example, channel 1 and channel 2) in the strategy that is being presented. This cost could change depending on a number of factors including the availability of channels, the interference levels, or the quality of service that is required. It is possible that the mathematical model, which is not presented in its whole, specifies a relationship or formula to calculate the channel allocation cost. It could take into account a wide range of relevant parameters and variables, such as the number of channels, channel characteristics, network conditions, or usage patterns, for example.

*Representation of Channel Allocation Cost:* The variables C1, C2, and so on represent the channel allocation cost for each specific channel (e.g., channel 1, channel 2, etc.). These variables are used to assign numerical values to the cost associated with using each channel within the mathematical model. Moreover, by using the mathematical model and the assigned channel allocation cost values, the equation allows for the calculation of the overall cost of channel allocation in the proposed approach. The specific form of the equation would depend on the details of the mathematical model, which are not provided in the given.

## 4. Model Description

Let us assume that we have *N* channels available for allocation. For each channel *i*, let $C_{i}$ represent the cost associated with allocating that channel. Additionally, let $X_{i}$ be a binary decision variable that takes the value 1 if channel *i* is allocated, and 0 otherwise.

The objective is to minimize the total channel allocation cost, given by
(17)$$\min\sum_{i=1}^{N}C_{i}\cdot X_{i}$$

Subject to the following constraints:(18)$$\begin{array}{cc} \mathrm{Channel}\;\mathrm{Allocation}\;\mathrm{Constraint}:\; & \sum_{i=1}^{N}X_{i}=K \end{array}$$(19)$$\begin{array}{cc} \mathrm{Non}\text{-}\mathrm{Negativity}\;\mathrm{Constraint}:\; & X_{i}\geq 0\;\forall i=1,2,\ldots ,N \end{array}$$

In Equation (2), *K* represents the total number of channels to be allocated, which should be specified based on the problem context. This mathematical model provides a representation of the channel allocation cost problem and can be used as a basis for further analysis and optimization.

### 4.1. Resource Optimization Approach

The following proposed mathematical model describes the resource optimization approach of the proposed framework. The resource that we consider here is the blockchain storage (memory).
(20)$$L_{\pi }\left(\theta \right)=\log\pi \left(u_{k}^{\left(t\right)}|x_{k}^{\left(t\right)};\theta \right)\Phi \left(x_{k},u_{k};\theta ,\theta_{v}\right).$$
(21)$$L_{v}\left(\theta_{v}\right)=\Phi \left(x_{k},u_{k};\theta ,\theta_{v}\right).$$
(22)$$\nabla_{\theta }L_{\pi }\left(\theta \right)=\nabla_{\theta }\log\pi \left(u_{k}^{\left(t\right)}|x_{k}^{\left(t\right)};\theta \right)\Phi \left(x_{k},u_{k};\theta ,\theta_{v}\right).$$
(23)$$\nabla_{\theta }L_{v}\left(\theta_{v}\right)=\frac{\partial \Phi {\left(x_{k},u_{k};\theta ,\theta_{v}\right)}^{2}}{\partial \theta_{v}}.$$

In the above equation, $s_{k}$ denotes the session for access and communication. Moreover, *t* denotes the time taken during the session allocation and session duration. Here, $a_{k}$ denotes the allocation at for *k*th session and *t* time.

#### 4.1.1. Homomorphic Encryption

It is possible to perform computations on data that have been encrypted using a technique known as homomorphic encryption. This technique eliminates the need to first decode the data before proceeding with the computations. In other words, it allows computations to be carried out directly on the encrypted data, which results in the generation of an encrypted result that, when decrypted, corresponds to the result of the computations performed on the original unencrypted data. This allows the encrypted result to be compared to the result of the computations carried out on the unencrypted data. To put it another way, it is feasible to conduct computations on the data that have been encrypted. Homomorphic encryption is a type of encryption that enables calculations to be carried out on data that has been encrypted without necessitating access to the decryption key. This enables homomorphic encryption to be used in situations when the decryption key is not readily available. This method of encryption is gaining more and more favor as time goes on. In other words, it makes it feasible to perform operations on encrypted data, which results in an encrypted output that, once decrypted, is similar to what would have been received if the computations had been carried out on the unencrypted version of the original data. This makes it possible to carry out operations without compromising the security of the data. The concept of homomorphic encryption was initially described for the first time in 1978 by Rivest, Adleman, and Dertouzos. Having said that, the actual applications of this concept have only been developed in more recent times. Homomorphic encryption techniques frequently use mathematical approaches as the basis for their construction, including lattice-based cryptography and the application of elliptic curves. The utilization of homomorphic encryption is applicable to a wide range of scenarios and provides a variety of benefits. The capacity to outsource computing labor to a third party while maintaining the secrecy of data is one of the most significant advantages offered by this technology. This comes in especially handy in circumstances in which sensitive data need to be processed on the cloud in a manner that does not involve the contents of that data being divulged to anyone. The data are encrypted before being sent to the cloud so that the service provider can perform computations using the encrypted data after they have been sent. After first obtaining the results in an encrypted format from the service provider, the user can then obtain the encrypted results and decode them on their local device. The kinds of computations that are supported by various homomorphic encryption methods are used to categorize the various homomorphic encryption techniques that are available. The use of fully homomorphic encryption, also known as FHE, enables arbitrary calculations to be carried out on data that have been encrypted. These computations can include addition, multiplication, and even more sophisticated operations. FHE techniques, on the other hand, have a tendency to be computationally demanding and call for a large amount of processing resources. On the other hand, partially homomorphic encryption methods are limited to supporting particular kinds of computations, such as addition or multiplication, but not both at the same time. These techniques are, on the whole, more efficient than FHE while still providing functionally effective solutions for a variety of applications. It is essential to highlight that homomorphic encryption is a topic of ongoing research; yet, despite the fact that advancements have been made in this field in recent years, there are still restrictions and difficulties associated with its practical application. Performing computations on encrypted data is often more slower and needs more resources compared with working on data that have not been encrypted. This is one of the key issues, along with performance and computational overhead. However, continuing research and development aim to overcome these constraints and make homomorphic encryption more applicable to applications used in the real world.

There are numerous homomorphic encryption algorithms available, including partially homomorphic encryption and fully homomorphic encryption:

1. Partially Homomorphic Encryption: This form of homomorphic encryption permits calculations on encrypted data, albeit only for certain types of operations. For instance, it might facilitate either addition or multiplication tasks on encrypted data, but not a combination of both. 2. FHE is the most potent form of homomorphic encryption. It facilitates computations on encrypted data for operations involving both addition and multiplication, among others. Using FHE, intricate calculations can be conducted on encrypted data, all the while ensuring its privacy.

The use of homomorphic encryption has substantial repercussions for both individuals’ right to privacy and the integrity of their data in a variety of settings. For instance, it can be implemented in safe cloud computing environments, in which sensitive data are kept in the cloud and computations can be carried out on encrypted data without disclosing the original contents to the cloud service provider. In addition to this, it can be utilized in secure data analysis, the application of machine learning to encrypted data, and safe multi-party computation [38].

However, it is worth noting that homomorphic encryption is still an active area of research, and practical implementations face challenges such as computational efficiency and performance. While progress has been made, fully homomorphic encryption schemes are still relatively slow and computationally intensive compared with traditional encryption schemes. Nonetheless, ongoing research and development efforts aim to improve the efficiency and practicality of homomorphic encryption techniques [39].

#### 4.1.2. Main Objectives

The main objectives of this paper are listed below:To investigate the feasibility and effectiveness of homomorphic encryption in preserving privacy and ensuring data confidentiality in the context of IoT healthcare applications;To design and develop the HealthLock framework, integrating blockchain technology and homomorphic encryption to provide enhanced privacy preservation in IoT healthcare applications;To conduct a comparative analysis with existing privacy preservation approaches in IoT healthcare applications, assessing the advantages and disadvantages of the HealthLock framework;To assess the usability and user acceptance of the HealthLock framework through user studies and feedback from healthcare professionals and patients;To validate the security of the HealthLock framework by conducting vulnerability analysis, threat modeling, and penetration testing to identify potential vulnerabilities and to propose necessary countermeasures.

## 5. Problem Statement

Concerns have been expressed about the privacy and security of sensitive medical data as a result of the growing use of Internet of Things (IoT) devices in healthcare applications. The conventional methods of data storage and administration in this field are not sufficiently equipped to meet the ever increasing demands placed on the protection of personal information. The current healthcare system does not have a reliable method to protect the privacy of patients and to maintain the data’s integrity, nor does it have a system that can process and analyze the data quickly and effectively. In addition, standard methods of data encryption frequently require data decryption for computations, which leaves sensitive information vulnerable to the possibility of security breaches. There is a need for a safe and privacy-preserving solution that can protect patient data in IoT-based healthcare applications in order to overcome these restrictions and to meet the demands of the industry. With the help of this solution, authorized parties should be able to process and analyze sensitive medical data without jeopardizing patients’ right to privacy. In addition, the combination of blockchain technology and homomorphic encryption methods presents a potentially fruitful strategy for overcoming these obstacles. The distributed ledger technology known as blockchain offers a framework for the storage of data that is both irreversible and unchangeable, providing both transparency and resistance to tampering. The confidentiality of the data is maintained during the entirety of the calculation process by using homomorphic encryption, which enables direct computations to be performed on encrypted data. However, the design and implementation of a practical and efficient framework that combines blockchain technology and homomorphic encryption in IoT-based healthcare applications present several challenges. These challenges include the following:Scalability: Healthcare systems generate a massive volume of data that needs to be securely stored and processed. Ensuring the scalability of the HealthLock framework to handle the increasing amount of data while preserving privacy is a significant challenge.Performance: The framework must provide efficient data processing and analysis capabilities, considering the computational overhead introduced by homomorphic encryption. It should strike a balance between privacy preservation and processing efficiency to enable real-time or near-real-time healthcare applications.Integration with Existing Infrastructure: Healthcare systems already have established infrastructure and legacy systems. Integrating the HealthLock framework with these existing systems seamlessly and ensuring interoperability can be a complex task.User Acceptance: The success of the HealthLock framework relies on its acceptance and adoption by healthcare professionals, patients, and other stakeholders. Ensuring a user-friendly interface, addressing usability concerns, and providing proper training and support are crucial for its successful implementation.Regulatory Compliance: Healthcare data are subject to strict regulations and compliance requirements, such as HIPAA (Health Insurance Portability and Accountability Act). The HealthLock framework must adhere to these regulations and ensure compliance to maintain trust and legality.

In light of these challenges, there is a need to develop a comprehensive framework, HealthLock, that utilizes blockchain technology and homomorphic encryption to preserve privacy in IoT-based healthcare applications. This framework should address scalability, performance, integration, user acceptance, and regulatory compliance to provide a secure and privacy-preserving environment for healthcare data management.

## 6. Proposed Framework

HealthLock is a framework designed to address privacy concerns in IoT healthcare applications by leveraging blockchain technology and homomorphic encryption. It aims to provide secure and privacy-preserving storage, sharing, and analysis of sensitive healthcare data. This framework ensures confidentiality, integrity, and auditability while allowing authorized entities to perform computations on encrypted data. Figure 1 presents the working of the proposed framework.

### 6.1. System Architecture

The HealthLock framework consists of the following components:IoT Devices: These devices collect healthcare data from patients, such as vital signs, medical records, and sensor readings. They securely transmit the data to the HealthLock system.HealthLock Blockchain: The blockchain forms the core of the framework, providing a decentralized and immutable ledger for storing encrypted healthcare data and transaction records. It ensures data integrity, transparency, and auditability. The blockchain can be implemented using existing platforms like Ethereum or Hyperledger.HealthLock Smart Contracts: Smart contracts are deployed on the blockchain to enforce predefined rules and to automate transactions. They handle access control, data sharing permissions, and cryptographic operations.Homomorphic Encryption Module: This module enables computations on encrypted data without decrypting it. It ensures privacy by allowing authorized parties to perform computations on the encrypted healthcare data, preserving confidentiality while obtaining useful insights. Advanced homomorphic encryption schemes like partially homomorphic encryption (PHE) or fully homomorphic encryption (FHE) can be utilized.Data Access Control: HealthLock employs a robust access control mechanism to ensure that only authorized entities can access specific healthcare data. Access control policies are enforced through smart contracts and encryption techniques.Data Lifecycle Management: The framework manages the lifecycle of healthcare data through the following stages:Data Collection: IoT devices securely collect healthcare data from patients. These data are encrypted using appropriate homomorphic encryption schemes before transmission.Data Encryption: The collected healthcare data are encrypted using homomorphic encryption techniques, ensuring confidentiality.Data Storage: The encrypted healthcare data are stored in the HealthLock blockchain. Each data entry is associated with a unique identifier, patient information, and access control policies defined by smart contracts.Data Sharing: Authorized entities, such as healthcare providers or researchers, can request access to specific healthcare data. Access requests are validated through smart contracts, ensuring compliance with privacy policies.Computation on Encrypted Data: Authorized entities can perform computations on the encrypted data without decrypting it, utilizing the homomorphic encryption module. This allows for secure analysis and data processing while preserving privacy.Data Audit and Accountability: The blockchain’s transparent nature enables auditing of data access and modifications. Transaction records stored on the blockchain provide an immutable log of data interactions, ensuring accountability and transparency.Security and Privacy Considerations: HealthLock incorporates the following security and privacy measures:Data Encryption: Sensitive healthcare data are encrypted using homomorphic encryption, ensuring confidentiality.Access Control: Smart contracts enforce access control policies, allowing only authorized entities to access specific healthcare data.Secure Communication: Data transmission between IoT devices and the HealthLock system is secured using cryptographic protocols, preventing unauthorized access.Immutable Ledger: The blockchain provides an immutable and tamper-proof ledger, ensuring data integrity and preventing unauthorized modifications.Privacy-Preserving Computation: Homomorphic encryption allows authorized parties to perform computations on encrypted data without compromising privacy.

The proposed system architecture is presented in Figure 2.

Figure 3 presents the flowchart of the proposed access control and mining transaction. Figure 2 reveals that if 80% of the users agree on specifically mining the transaction, then it is approved; otherwise, access is denied.

The comprehensive evaluation of the proposed HealthLock system encompasses quantitative and qualitative metrics to assess its performance, privacy preservation capabilities, and efficiency. The evaluation aims to provide a thorough understanding of the system’s effectiveness and its potential for real-world deployment in IoT healthcare applications. Here are some metrics that can be considered for evaluating HealthLock:


*1. Privacy Preservation:*
-Confidentiality: Measure the ability of HealthLock to protect sensitive healthcare data from unauthorized access.-Anonymity: Evaluate the level of anonymity provided by HealthLock to ensure the privacy of individuals’ identities and health information.-Data Integrity: Assess the system’s ability to detect and prevent tampering or unauthorized modifications of healthcare data stored on the blockchain.



*2. Performance Metrics:*
-Throughput: Measure the number of transactions or data requests that HealthLock can handle per unit of time.-Latency: Evaluate the time taken for data retrieval, storage, and processing within the HealthLock system.-Scalability: Assess the system’s ability to handle a growing number of IoT devices, healthcare providers, and data transactions while maintaining performance.



*3. Efficiency Metrics:*
-Computational Overhead: Measure the computational resources required for the execution of cryptographic operations, such as homomorphic encryption and blockchain consensus algorithms.-Energy Consumption: Evaluate the energy efficiency of the HealthLock system, especially in resource-constrained IoT devices.-Storage Requirements: Assess the amount of storage space required by the HealthLock blockchain to store encrypted healthcare data.



*4. Security Evaluation:*
-Vulnerability Analysis: Conduct a thorough assessment of the system’s security vulnerabilities, including potential attacks on the blockchain, encryption schemes, and smart contracts.-Penetration Testing: Perform controlled testing to identify any weaknesses or vulnerabilities in the HealthLock system’s infrastructure.-Threat Modeling: Examine potential hazards and vulnerabilities that could impact the privacy and security of healthcare information within the HealthLock system.



*5. User Acceptance and Usability:*
-User Satisfaction: Gather feedback from healthcare professionals, patients, and other stakeholders to evaluate their satisfaction with the HealthLock system’s privacy features and usability.-User Experience: Assess the ease of use and user-friendliness of the HealthLock system’s interface and functionalities.



*6. Comparative Analysis:*
-Compare the performance, privacy preservation capabilities, and efficiency of HealthLock with existing privacy preservation approaches in IoT healthcare applications.-Evaluate the advantages and disadvantages of HealthLock in comparison with other state-of-the-art solutions.


By conducting a comprehensive evaluation using these metrics, the proposed HealthLock system can be rigorously assessed for its effectiveness in preserving privacy, performance, and efficiency in IoT healthcare applications. The results of the evaluation will provide valuable insights into the system’s strengths, limitations, and potential areas for improvement, ultimately validating its viability as a privacy-preserving solution in the healthcare domain [40].

### 6.2. Analysis of Correctness and Unforgeability of HE

Correctness, unconditional anonymity, and the inability to be forged are the three requirements that need to be satisfied by a safe homomorphic encryption system. To be able to build a mathematical model for the examination of the correctness and unforgeability of a homomorphic encryption (HE) scheme, we first need to specify the formal framework and notation for the encryption and decryption operations [41]. Only then can we begin writing the model. The generic partially homomorphic encryption system that allows at least one homomorphic operation, such as addition or multiplication, will be the primary focus of our attention here. Let us define the notations:

M is the set of all possible plaintext messages.

-Ciphertext space: C is the set of all possible ciphertexts.-Key space: K is the set of all possible encryption and decryption keys.-Key generation algorithm: KeyGen is the algorithm that generates the encryption and decryption keys, i.e., $\left(pk,sk\right)\leftarrow \mathrm{KeyGen}\left(1^{\lambda }\right)$, where pk is the public key and sk is the secret key, and λ represents the security parameter.-Encryption algorithm: Enc refers to the encryption algorithm which, when given a public key pk and a plaintext message m from set M, results in a ciphertext c∈C, i.e., $c\leftarrow {\mathrm{Enc}}_{pk}\left(m\right)$.-Decryption algorithm: Dec is the decryption algorithm that takes the secret key sk and a ciphertext c∈C and recovers the original plaintext message m∈M, i.e., $m\leftarrow {\mathrm{Dec}}_{sk}\left(c\right)$.-Homomorphic operation: Let ★ be the homomorphic operation supported by the HE scheme. Given ciphertexts $c_{1}$ and $c_{2}$, we can compute $c_{3}=c_{1}★c_{2}$, and after decryption, we can obtain $m_{3}=m_{1}★m_{2}$, where $m_{1}$ and $m_{2}$ are the plaintext messages corresponding to $c_{1}$ and $c_{2}$, respectively.

**Correctness Property:** The correctness of an HE scheme implies that after performing homomorphic operations, the result of decryption matches the corresponding operation performed on plaintext messages. In other words, if $c_{1}={\mathrm{Enc}}_{pk}\left(m_{1}\right)$ and $c_{2}={\mathrm{Enc}}_{pk}\left(m_{2}\right)$, then after computing $c_{3}=c_{1}★c_{2}$ and decrypting it with the secret key, we should obtain $m_{3}=m_{1}★m_{2}$.

Mathematically, the correctness property can be expressed as follows:
(24)$$\left[m_{1},m_{2}\in M:{\mathrm{Dec}}_{sk}\left({\mathrm{Enc}}_{pk}\left(m_{1}\right)★{\mathrm{Enc}}_{pk}\left(m_{2}\right)\right)=m_{1}★m_{2}.\right]$$

**Unforgeability Property:** The unforgeability property of a HE scheme ensures that an adversary cannot produce a valid ciphertext for a message that it has not encrypted before, even if they have access to the public key and the ability to perform valid homomorphic operations on ciphertexts.

Mathematically, the unforgeability property can be expressed using a game between the adversary A and a challenger:

1. The challenger runs $\mathrm{KeyGen}(1^{\lambda })$ to generate the public key pk and the secret key sk. 2. The adversary A has access to the public key pk and can submit ciphertexts for decryption. 3. The adversary A can perform homomorphic operations on ciphertexts and submit the result for decryption. 4. The adversary A wins the game if they can produce a valid ciphertext *c* that decrypts to a meaningful message *m* without having obtained *c* through ${\mathrm{Enc}}_{pk}\left(\cdot \right)$.

The HE scheme is considered unforgeable if for all efficient adversaries A,
(25)$$\left[\mathrm{Pr}\left[\begin{array}{cc} & \left(pk,sk\right)\leftarrow \mathrm{KeyGen}\left(1^{\lambda }\right),\\ & m\leftarrow A(pk,{\mathrm{Enc}}_{pk}\left(\cdot \right)):\\ & {\mathrm{Dec}}_{sk}\left(c\right)=m\;\mathrm{and}\;c\notin {\mathrm{Enc}}_{pk}\left(M\right) \end{array}\right]\leq \mathrm{negl}\left(\lambda \right).\right]$$

Here, ${\mathrm{Enc}}_{pk}\left(M\right)$ represents the set of ciphertexts obtained by encrypting all possible messages in the plaintext space M.

The above model provides a general framework for understanding the correctness and unforgeability properties of a partially homomorphic encryption scheme.

### 6.3. Proposed Algorithm

This section provides the proposed algorithm integrated with our proposed framework. These algorithm are explained one by one below: The proposed Algorithm 1 outlines the steps for adding a node to the blockchain network based on the present paper. It begins by initializing the necessary variables, including the new node’s identifier and public key. The process begins when a new node submits a join request to the network. If this request is accepted, a new block is generated and includes pertinent data like the hash of the preceding block, timestamp, nonce, merkle root, and the ID and public key of the node.
**Algorithm** **1** Add node to blockchain network. **Input**: NewNode **Output**: UpdatedBlockchain **Initialization**: *nodeID*←NewNode’s unique identifier; *nodePublicKey*←NewNode’s public key; *blockchain*←Existing blockchain network; **Procedure**: *NewNode* sends a request to join the network; Request is approved *newBlock*←CreateGenesisBlock(); *newBlock.header.previousHash*←blockchain.getLatestBlock().header.blockHash; *newBlock.header.timestamp*←CurrentTimestamp(); *newBlock.header.nonce*←CalculateNonce(); *newBlock.header.merkleRoot*←CalculateMerkleRoot(); *newBlock.header.nodeID*←nodeID; *newBlock.header.nodePublicKey*←nodePublicKey; *newBlock.header.blockHash*←CalculateBlockHash(newBlock.header); *blockchain.addBlock(newBlock)* **Broadcast** the new block to the network;  **Update** the blockchain network; **Return** UpdatedBlockchain

Algorithm 2 outlines the steps for searching health records based on a given search query. Moreover, our proposed Algorithm 2 takes the search query and the blockchain as input and returns a list of matching records as output. The procedure iterates over each block in the blockchain and checks each record within the block. If a record matches the search query, it is added to the list of matching records. Finally, the algorithm returns the list of matching records [42].
**Algorithm** **2** Health record search. **Input**: SearchQuery, Blockchain **Output**: MatchingRecords **Initialization**:;  *MatchingRecords*←Empty list; **Procedure**:; block in Blockchain record in block record matches SearchQuery *MatchingRecords.append(record)* **Return** MatchingRecords

The proposed Algorithm 3 outlines the steps for security and privacy preservation against DoS, ransomware, and phishing attacks. It takes an incoming request as input and returns the appropriate response as output. The procedure begins by checking if the incoming request is a DoS attack. If it is, the algorithm executes the “BlockRequest” function to block the request and to prevent the DoS attack. If the request is determined to be a ransomware attack, the algorithm isolates the infected system through the “IsolateInfectedSystem” function. In the case of a phishing attack, the algorithm verifies the identity of the request through the “VerifyIdentity” function. For any other requests, the algorithm processes them normally through the “ProcessRequest” function. Finally, the algorithm returns the appropriate response based on the type of attack or request.
**Algorithm** **3** Security and privacy preservation. **Input**: IncomingRequest **Output**: Response **Initialization**:; *Response*←Empty; **Procedure**:; IncomingRequest is a DoS attack *Response*←BlockRequest(); IncomingRequest is a ransomware attack *Response*←IsolateInfectedSystem(); IncomingRequest is a phishing attack *Response*←VerifyIdentity(); *Response*←ProcessRequest(); **Return** Response;

#### Solidity Tool

An event on Ethereum is a well-defined way of asynchronously transmitting data among members of the network. Smart contracts can be triggered by events in Solidity, which are delivered as signals. Ethereum JSON-RPC API (a blockchain network interface offered for connectivity and programmatic interactions with blockchain) allows DApps, which are effectively decentralized apps, the ability to listen to these events and to act accordingly [42]. It is also possible to index an event, so that the event’s history may be found later.

## 7. Simulation Setup

In order to implement the proposed approach, the simulation setup for the experiment was set as follows:Cache System Configuration: Determine the size and capacity of the cache system, including the number of cache slots or cache memory available for storing data items. Moreover, choose a cache replacement policy, such as Least Recently Used (LRU), Least Frequently Used (LFU), or Random, to manage cache eviction and replacement.Data Access Pattern: Generate or simulate a workload that represents the data access pattern in the system. This step consists of a series of requests for specific data items or a stream of data access events [42].Cache Hit and Miss Determination: Based on the data access pattern, determine whether each access results in a cache hit (the data item is found in the cache) or a cache miss (the data item is not found in the cache and needs to be retrieved from a slower storage system).Search Time Measurement: Measure the time taken to search for a data item in the cache, including the time required to check cache slots, to compare keys, and to retrieve the requested data.Cache Hit Rate Calculation: Compute the cache hit rate by determining the ratio of cache hits to the total cache interactions (the sum of hits and misses), expressed as a percentage.

## 8. Results

In this section we provide simulation results based on the experimental setup. By simulating different scenarios with varying numbers of domains and analyzing the relationship between the number of domains and cache hit rate, the following general trends might be observed through Figure 4:

Impact of Domain Diversity: As the number of domains increases, the cache system experiences a wider range of requests and data access patterns. This increased diversity can lead to a lower overall cache hit rate because the cache system needs to accommodate a larger variety of data items from different domains.

Cache Size and Capacity: The cache hit rate can be influenced by the size and capacity of the cache relative to the number of domains. If the cache size is insufficient to store data items from all domains effectively, the cache hit rate may decrease as the number of domains increases. Increasing the cache size or implementing dynamic cache resizing strategies can help mitigate this effect.

Cache Replacement Policy: The choice of cache replacement policy can impact the cache hit rate for different domains. Some domains may have more temporal locality, where recently accessed data items are likely to be accessed again, while others may exhibit different access patterns. Adapting the cache replacement policy to suit the characteristics of each domain can improve the cache hit rate.

Inter-Domain Data Correlations: The cache hit rate may vary based on the presence of correlations or overlaps in data accessed across domains. If multiple domains frequently access the same data items, the cache hit rate can benefit from caching these shared items. Conversely, if domains have mostly distinct data access patterns, the cache hit rate may decrease as the number of domains increases [43].

Cache Preloading Strategies: Preloading the cache with frequently accessed data items from each domain or using domain-specific caching strategies can help improve the cache hit rate. By understanding the access patterns and data dependencies between domains, customizing cache initialization techniques can enhance the overall cache effectiveness. Figure 5 represent the simulation results based on the Comparison of the encryption time between the proposed approach and related works with respect to the transactions number.

By varying the size of the dataset and performing multiple simulations, the following general trends might be observed through Figure 6:

Increasing Data Size and Accuracy: As the dataset size increases, the model may have access to more diverse and representative examples, which can improve its learning ability. Consequently, the accuracy of the model on the testing data is expected to increase, up to a certain point. As a result, we anticipate that the model’s accuracy on the test data will improve, though only to a certain limit. At a certain point, adding more data may result in diminishing returns regarding accuracy improvement. This is because the model might already have learned the underlying patterns in the data and additional examples may not significantly contribute to further improvement. In some cases, when the data-set size is small, the model might exhibit over-fitting, meaning it performs exceptionally well on the training data but poorly on the testing data. Increasing the data-set size can help alleviate over-fitting and improve generalization, leading to better accuracy on unseen data. It is essential that the connection between the quantity of data and its correctness might shift in nature based on the nature of the individual problem, the intricacy of the data-set, and the particular machine learning algorithms that are used. In addition, the results of the simulation must to be analyzed with caution and validated using data taken from the real world in order to arrive at trustworthy conclusions.

Figure 7 was calculated using the amount of time spent training and the anticipated number of violations. If the amount of time spent training is varied and several simulations are carried out, it is possible that the following broad patterns will emerge: The relationship between training time and detection accuracy in cybersecurity models is really quite complicated. It is dependent on a wide range of criteria, such as the model architecture, the size of the dataset, the quality of the data, and the type of cybersecurity breaches that are being identified.

1. **Diminishing Returns:** As training time increases, the model may start reaching a point of diminishing returns, where further exposure to data does not result in significant improvements in accuracy. At this stage, the model may have already learned the most important patterns, and additional training time may yield only marginal gains.

2. **Overfitting:** Prolonged training can lead to overfitting, where the model becomes too specific to the training data and loses its ability to generalize to new, unseen data. Overfit models may perform exceptionally well on the training data but fail to perform well on real-world data, leading to reduced accuracy in practical scenarios.

3. **Validation Set Monitoring:** To mitigate overfitting and to identify the optimal training time, it is essential to monitor the model’s performance on a validation set during the training process. The validation set serves as a proxy for unseen data, allowing us to assess how well the model generalizes to new examples. If the validation performance starts to degrade, it may indicate that the model is overfitting, and training should be stopped or adjusted.

4. **Regularization Techniques:** Regularization techniques, such as L1 and L2 regularization, dropout, or early stopping, can be employed to prevent overfitting and to improve generalization. These techniques penalize overly complex model parameters, making the model more resilient to noise in the training data.

5. **Model Complexity:** The complexity of the model architecture also influences the relationship between training time and accuracy. More complex models may require longer training times to converge and may be more prone to overfitting.

6. **Computational Resources:** Longer training times often require more computational resources, which may not always be feasible or cost-effective. Balancing training time with available resources is crucial in real-world applications.

In practice, it is essential to strike a balance between training time, accuracy, and generalization to achieve a cybersecurity model that performs well on both training and unseen data. Proper validation and regularization techniques can help optimize the model’s performance and prevent overfitting, ensuring its effectiveness in real-world cybersecurity scenarios.

By simulating different scenarios and analyzing the relationship between search time and cache hit rate, the following general trends might be observed through Figure 7:Search Time and Cache Hit Rate Trade-off: Generally, as the cache hit rate increases, the search time tends to decrease. This is because a higher cache hit rate implies a larger proportion of data being found in the cache, reducing the search time required to retrieve the requested items.Cache Size Impact: Increasing the cache size typically improves the cache hit rate, as it allows more data items to be stored in the cache. A larger cache size increases the likelihood of finding frequently accessed data items in the cache, leading to shorter search times.Cache Replacement Policy: The choice of cache replacement policy can impact both search time and cache hit rate. For example, LRU replacement tends to have a higher cache hit rate for workloads with temporal locality, while LFU replacement may better handle workloads with frequent access to a few specific items.Access Pattern Influence: The specific data access pattern in the simulation can affect the relationship between search time and cache hit rate. Workloads with high locality of reference, where the same data items are repeatedly accessed, tend to have higher cache hit rates and shorter search times. It is obvious that the proposed approach outperforms the existing model.

## 9. Conclusions

The aim of this article was to create a secure, blockchain-oriented framework for preserving data privacy in the healthcare sector, facilitated by the Internet of Things (IoT). The framework employs data encryption technologies to guarantee robust security. The techniques of deep learning have been further applied for the purpose of data prediction in the medical field. In the beginning, the OK-HECCFHE that had been developed was used to help in the gathering and encrypting of the medical data. During the ECC-based encryption phase, the ideal group key was chosen for the purpose of lowering the amount of required computing time and memory space. The data had been encrypted before being saved in the blockchain database, which ensured its safety. The decryption was carried out only when it became necessary to access these previously encrypted medical data. In addition, these decrypted data were utilized within the context of the medical data prediction model. Within this model, the ODNN-GRU was developed with DNN and GRU characteristics with the intention of accurately predicting the medical data. This prediction performance was further improved with the model for optimizing the parameters in the prediction network with the goal of obtaining better accuracy and precision in the prediction outcomes. In the comparison of the suggested and traditional techniques, the analysis of the results revealed that the proposed techniques had greater degrees of accuracy, 7.52%, 5.37%, 3.11%, and 1.17%, respectively, when compared with the accuracy of the Merkle Tree, DNN, GRU, and DNN-GRU when using dataset 2. As a result, the blockchain-based secured medical data prediction model that was developed has offered outstanding performance in both the encryption of data and the prediction of medical data. In the future, we will focus on the following extension to our work:1Scalability and Performance Optimization: One avenue for future work is to explore methods for improving the scalability and performance of the HealthLock framework. This could involve optimizing the blockchain implementation to handle a larger volume of healthcare data and transactions efficiently. Techniques such as sharding, sidechains, or off-chain solutions can be investigated to enhance scalability while maintaining data privacy and security. Research and development of more advanced homomorphic encryption schemes can be pursued to improve the efficiency and functionality of the HealthLock framework. This might involve investigating partially homomorphic encryption (PHE) or fully homomorphic encryption (FHE) methods that allow for advanced calculations on encrypted data, while maintaining confidentiality. Investigating the use of pre-processing techniques, optimization algorithms, and hardware acceleration can also contribute to faster and more practical homomorphic encryption solutions. Moreover, Integrating privacy-preserving analytics and machine learning techniques within the HealthLock framework can be a promising avenue for future work. Developing methods to perform advanced computations, such as statistical analysis, predictive modeling, or anomaly detection, directly on encrypted data can enable valuable insights while maintaining privacy. Exploring techniques like secure multiparty computation (MPC) and federated learning can facilitate collaborative analyses on encrypted data without exposing sensitive information.

## 10. Future Work

2Real-World Implementation and Deployment: Conducting real-world pilot studies and deploying the HealthLock framework within healthcare organizations can provide valuable insights into its practical feasibility, performance, and user acceptance. Collaborating with healthcare providers, patients, and regulatory bodies can help address legal and ethical considerations, validate the framework’s effectiveness, and identify potential challenges and improvements in real-world settings.3Usability and User Experience: Investigating the usability and user experience aspects of the HealthLock framework is an essential area of future work. Designing intuitive user interfaces, considering user perspectives, and conducting user studies can contribute to the framework’s adoption and acceptance among healthcare professionals, patients, and other stakeholders. Incorporating user feedback and continuously refining the framework’s usability can enhance its practical implementation and ensure that it aligns with user requirements.4Security and Threat Analysis: Conducting a comprehensive security and threat analysis is crucial for ensuring the robustness of the HealthLock framework. Performing vulnerability assessments, penetration testing, and adversarial modeling can help identify potential weaknesses or attack vectors.

## Figures and Tables

**Figure 1 sensors-23-06762-f001:**
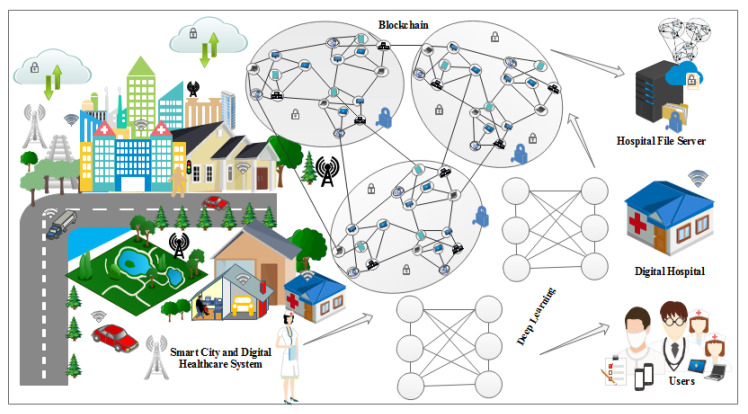
Proposed framework using homomorphic encryption integrated with hybrid deep learning.

**Figure 2 sensors-23-06762-f002:**
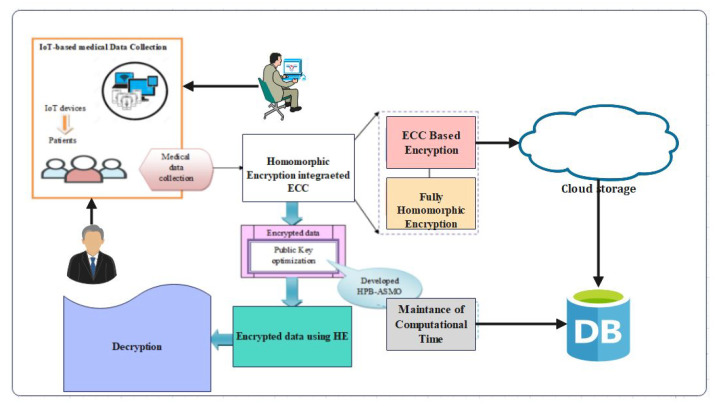
Proposed framework (HealthLock).

**Figure 3 sensors-23-06762-f003:**
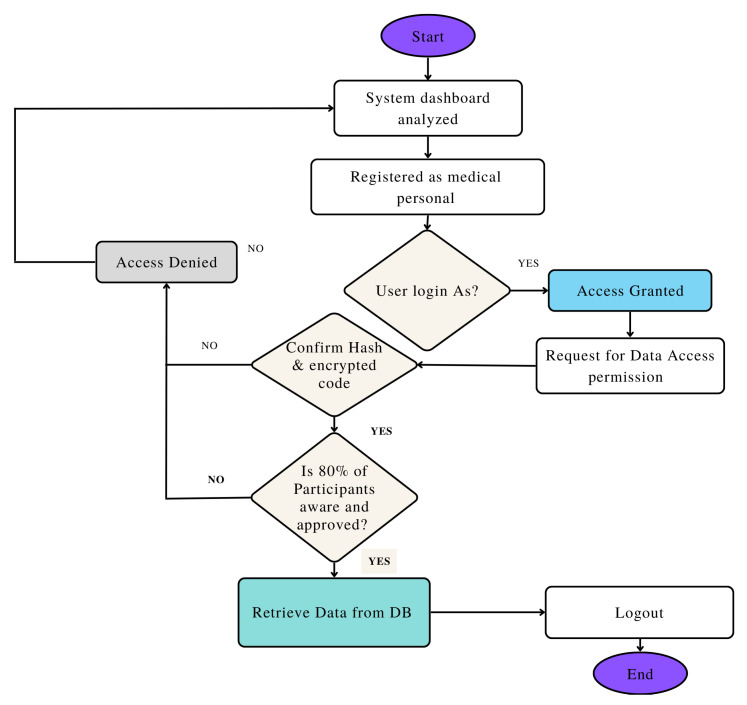
Proposed flowchart for authorization of users.

**Figure 4 sensors-23-06762-f004:**
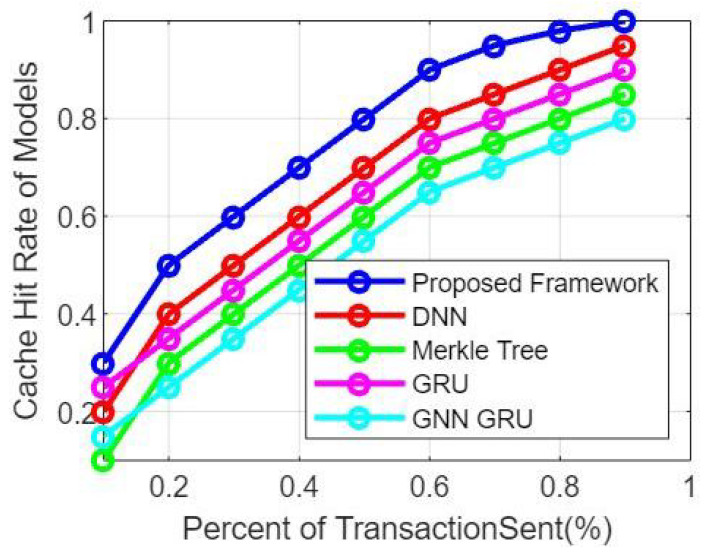
Comparison of the cache hit rate between the proposed framework and related works with respect to the percent of transactions.

**Figure 5 sensors-23-06762-f005:**
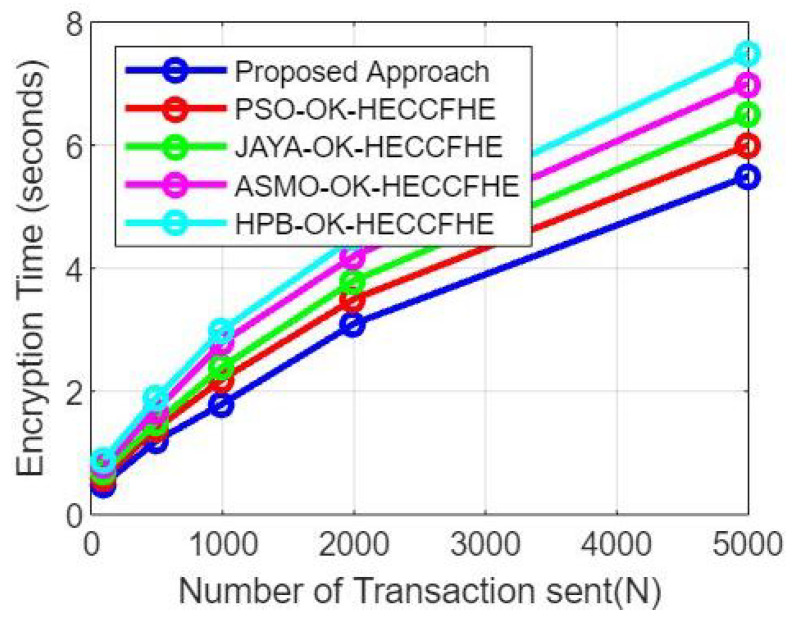
Comparison of the encryption time between the proposed approach and related works with respect to the transactions number.

**Figure 6 sensors-23-06762-f006:**
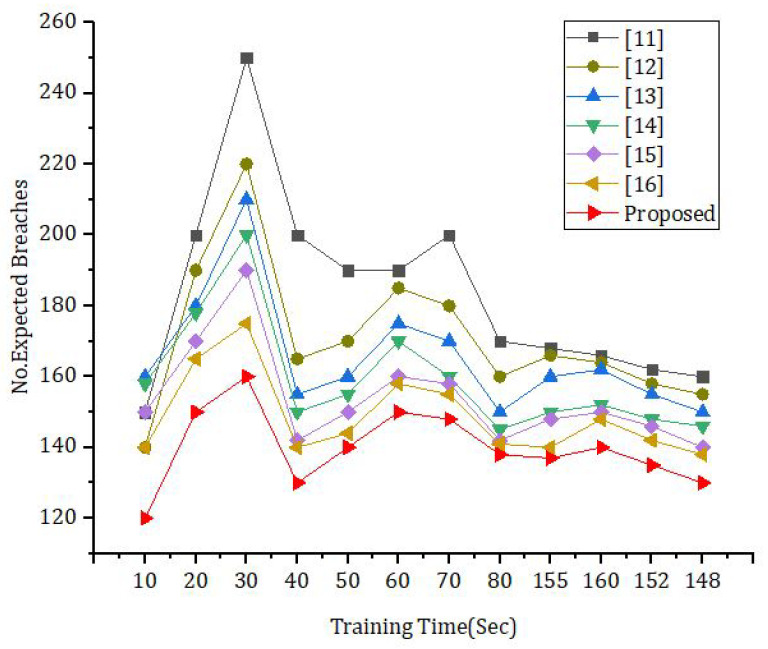
Simulation results based on training time and number of expected breaches.

**Figure 7 sensors-23-06762-f007:**
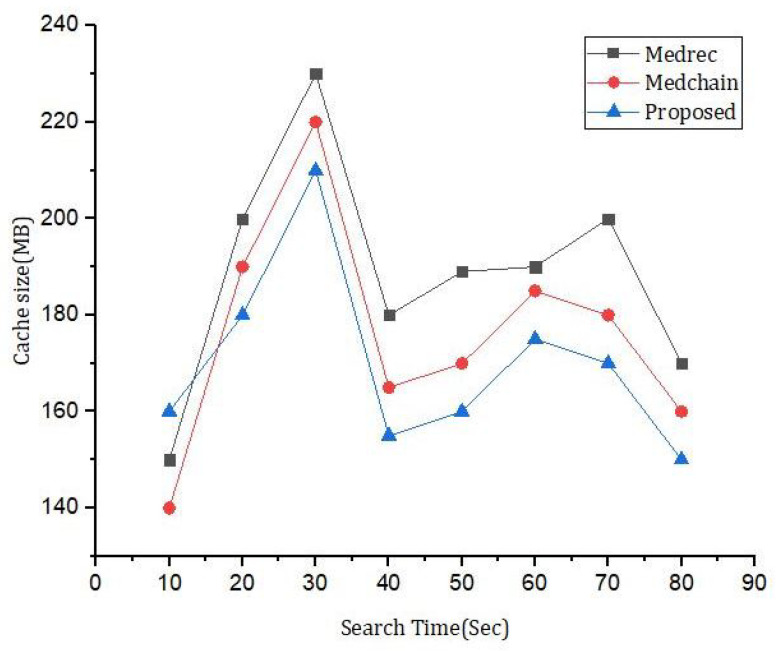
Comparison of cache size between the proposed framework and related work with respect to search time.

## Data Availability

The data will be provided upon request.

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
