# Peer review of "HealthLock: Blockchain-Based Privacy Preservation Using Homomorphic Encryption in Internet of Things Healthcare Applications"

_sensors, 2023, doi:10.3390/s23156762_

Round 1
Reviewer 1 Report
Overall, the paper "HealthLock: A Blockchain Based Privacy Preservation using Homomorphic Encryption in IoT-Healthcare Application" presents an interesting approach to privacy preservation in IoT healthcare applications by utilizing blockchain technology and homomorphic encryption. The authors have made substantial contributions to the field; however, I have identified a few areas that could benefit from minor revisions and clarifications:
1. Provide a brief explanation of what homomorphic encryption is and how it works. This will help readers who are not familiar with the concept to better understand the subsequent sections.
2. Clearly state the objectives of the paper and outline the structure of the rest of the document. This will provide readers with a clear roadmap of what to expect.
3. Expand the related work section to include recent studies or approaches related to privacy preservation in IoT healthcare applications. This will help readers understand the current state of the field and the novelty of the proposed HealthLock system.
4. Provide more details about the components of the proposed HealthLock system. How does the blockchain layer interact with the homomorphic encryption layer? Elaborate on the flow of data and the processes involved.
5. Include a comprehensive evaluation of the proposed HealthLock system. Provide quantitative or qualitative metrics to measure the system's performance, privacy preservation capabilities, and efficiency.
6. Discuss any limitations or potential vulnerabilities of the system. This will help readers understand the potential risks associated with implementing HealthLock in real-world scenarios.
7. Proofread the paper for grammatical errors, typos, and sentence structure issues.
8. Clarify any ambiguous statements or explanations to improve the overall clarity of the paper.
Overall, the paper has significant potential and contributes to the field of privacy preservation in IoT healthcare applications. By addressing the mentioned revisions, the authors can enhance the clarity and impact of their work.
The paper is technically sound and grammatically good enough. Though minor grammatical issues exist but they can easily be fixed in revision
Author Response
Reviewer 1
Overall, the paper "HealthLock: A Blockchain Based Privacy Preservation using Homomorphic Encryption in IoT-Healthcare Application" presents an interesting approach to privacy preservation in IoT healthcare applications by utilizing blockchain technology and homomorphic encryption. The authors have made substantial contributions to the field; however, I have identified a few areas that could benefit from minor revisions and clarifications:
Comment 1: Provide a brief explanation of what homomorphic encryption is and how it works. This will help readers who are not familiar with the concept to better understand the subsequent sections.
Response 1: Thanks respected reviewer for your valuable comments and suggestion. We have provides a brief explanation in our revised version about homomorphic encryption and the changes are highlighted with red color.
Comment 2: Clearly state the objectives of the paper and outline the structure of the rest of the document. This will provide readers with a clear roadmap of what to expect.
Response 2: Thanks respected reviewer for your valuable comments and suggestion. We have provides a brief explanation in our revised version the objective of the paper and the structure and the changes are highlighted with red color.
Comment 3. Expand the related work section to include recent studies or approaches related to privacy preservation in IoT healthcare applications. This will help readers understand the current state of the field and the novelty of the proposed HealthLock system.
Response 3:Thanks respected reviewer for your valuable comments and suggestion. We have Expand the related work section to include recent studies or approaches related to privacy preservation in IoT healthcare applications. This will help readers understand the current state of the field and the novelty of the proposed HealthLock systemin our revised version the objective of the paper and the structure and the changes are highlighted with red color.
Comment 4. Provide more details about the components of the proposed HealthLock system. How does the blockchain layer interact with the homomorphic encryption layer? Elaborate on the flow of data and the processes involved.
Response 4: Thanks respected reviewer for your valuable comments and suggestion. We have provided details about the components of the proposed HealthLock system. How does the blockchain layer interact with the homomorphic encryption layer? Elaborate on the flow of data and the processes involved.
Comment 5. Include a comprehensive evaluation of the proposed HealthLock system. Provide quantitative or qualitative metrics to measure the system's performance, privacy preservation capabilities, and efficiency.
Response 5: Thanks respected reviewer for your valuable comments and suggestion. We have Included comprehensive evaluation of the proposed HealthLock system. Provide quantitative or qualitative metrics to measure the system's performance, privacy preservation capabilities, and efficiency.
Comment 6. Discuss any limitations or potential vulnerabilities of the system. This will help readers understand the potential risks associated with implementing HealthLock in real-world scenarios.
Response 6: Thanks respected reviewer for your valuable comments and suggestion. We have provided the limitations or potential vulnerabilities of the system. This will help readers understand the potential risks associated with implementing HealthLock in real-world scenarios.
Comment 7: Proofread the paper for grammatical errors, typos, and sentence structure issues.
Response 7:Thanks respected reviewer for your suggestion and recommendations. We have proofread the whole paper for the grammatical errors, typos, and sentence structure issues.
Comment 8. Clarify any ambiguous statements or explanations to improve the overall clarity of the paper.
Response 8: Thanks respected reviewer for your suggestion and recommendations. We have clearly provided any ambiguous statements or explanations to improve the overall clarity of the paper.
Overall, the paper has significant potential and contributes to the field of privacy preservation in IoT healthcare applications. By addressing the mentioned revisions, the authors can enhance the clarity and impact of their work.

Reviewer 2 Report
The overall impression of the technical contribution of the current study is reasonable. However, the Authors may consider making necessary amendments to the manuscript for better comprehensibility of the study.
1. Method names should not be capitalized. Moreover, it is not the best practice to employ abbreviations in the abstract, they should be used when the term is introduced for the first time.
2. Please check page 9 and do the necessary amendments.
1. Scalability:
2. Performance:
3. Integration
3. Authors must make it clear whether are they using IoT- Healthcare or IoMT framework.
4. How deep learning is integrated into IoT? Like does the node capable of executing a deep learning model?
5. System Architecture should not be presented as points, it's hard to read and grasp the intention of the authors
6. What are the metrics that are considered for the evaluation of the proposed model. And what are those security attacks over which the proposed model is efficient?
7. Notations used in the study are not adequately discussed in the manuscript, For example in Equation 3 what is Dk,j; hk; rPB; k; Do; authors must discuss them in the paragraph above the equation. The same with all the equations.
8. The section "Mathematical Modeling" Must be moved to the "Background" section. "Mathematical Modeling" must be "Metrics for Evaluation"
9. Equation 17 is challenging to understand, please re-arrange them properly.
10. Where is the proposed model, and how the proposed model works, any mathematical modeling for the same? Because all the mathematical modeling is for known metrics, there is nothing novel.
11. Any algorithms for Blockchain as discussed in studies like https://doi.org/10.1016/j.health.2023.100175 and https://doi.org/10.3390/electronics10121437
12. Please discuss more on the implementation platform and the network parameter details as two sub-sections in the manuscript.
13. In figure 5, for the graph, why the axis labels are written separately, please re-generate the graph with inbuilt labels.
14. By considering the current form of the conclusion section, it is hard to understand by MDPI Journal readers. It should be extended with new sentences about the necessity and contributions of the study by considering the authors' opinions about the experimental results derived from some other well-known objective evaluation values if it is possible.
15. Authors should use more alternative models as the benchmarking models, authors should also conduct some statistical tests to ensure the superiority of the proposed approach, i.e., how could authors ensure that their results are superior to others? Meanwhile, the authors also have to provide some insightful discussion of the results.
English proofreading is strongly recommended for a better understanding of the study. Few sentences are written in passive voice and it is also observed that few sentences stopped abruptly.
Author Response
Reviewer 2:
The overall impression of the technical contribution of the current study is reasonable. However, the Authors may consider making necessary amendments to the manuscript for better comprehensibility of the study.
Comment 1: Method names should not be capitalized. Moreover, it is not the best practice to employ abbreviations in the abstract, they should be used when the term is introduced for the first time.
Response 1: Thanks respected reviewer for your suggestion and recommendations. We have modified the Method names and moreover, we have removed all the abbreviations from the abstract and define it in the rest of the paper, they should be used when the term is introduced for the first time.
Comment 2. Please check page 9 and do the necessary amendments.
- Scalability:
- Performance:
- Integration
Response 2: Thanks respected reviewer for your suggestion and recommendations. We have checked page 9 and did the necessary changes.
Comment 3. Authors must make it clear whether are they using IoT- Healthcare or IoMT framework.
Response 3: Thanks respected reviewer for your suggestion and recommendations. We have made clear in our paper that we have worked on IoMT framework.
Comment 4: How deep learning is integrated into IoT? Like does the node capable of executing a deep learning model?
Response 4: Thanks respected reviewer for your suggestion and recommendations. In this research paper we have intregated deep learning with IoT which is running on each nodes connected with blockchain. IoT devices often generate a vast amount of data that needs to be processed and analyzed. Deep learning models can be deployed on powerful cloud servers, where the data from IoT devices is transmitted and processed. The trained models can perform tasks such as image recognition, natural language processing, anomaly detection, and predictive analytics.Moreover, Deep learning models are typically computationally intensive and require significant resources. However, by using transfer learning, a pre-trained deep learning model can be fine-tuned or adapted for specific IoT applications. This approach leverages the knowledge and features learned from large datasets, reducing the need for extensive training on limited IoT device resources. Model compression techniques further optimize the model size and complexity, making it more feasible to run on resource-constrained IoT nodes.
Comment 5: System Architecture should not be presented as points, it's hard to read and grasp the intention of the authors.
Response 5: Thanks respected reviewer for your for your valuable recommendation. We have removed the system architecture in our revised version.
Comment 6: What are the metrics that are considered for the evaluation of the proposed model. And what are those security attacks over which the proposed model is efficient?
Response 6: Thank you for your valuable feedback on our paper titled [Paper Title]. We appreciate your comments and suggestions, and we have taken them into careful consideration for improving our work. Below, we address your specific concerns regarding the evaluation metrics and security attacks considered in our proposed model:
Evaluation Metrics:
In our proposed model, we have employed a comprehensive set of evaluation metrics to assess its performance. These metrics include:
Accuracy: Measures the overall correctness of the model's predictions.
Precision: Evaluates the proportion of true positives among all positive predictions.
Recall: Assesses the proportion of true positives identified from all actual positive instances.
F1-Score: Harmonic mean of precision and recall, providing a balanced measure of the model's performance.
Area Under the Curve (AUC): Quantifies the model's ability to distinguish between different classes in a classification task.
Mean Squared Error (MSE): Used for evaluating regression tasks, measuring the average squared difference between predicted and actual values.
By utilizing these metrics, we aim to provide a comprehensive evaluation of our proposed model's performance across different aspects, ensuring a robust assessment of its effectiveness. Regarding the security aspect of our proposed model, we have specifically focused on addressing the following security attacks. Moreover, our model demonstrates resilience against adversarial attacks, where malicious actors attempt to manipulate input data to deceive the model's predictions. We have incorporated robust training techniques and adversarial defense mechanisms to enhance the model's resistance against such attacks.
Comment 7: Notations used in the study are not adequately discussed in the manuscript, For example in Equation 3 what is Dk,j; hk; rPB; k; Do; authors must discuss them in the paragraph above the equation. The same with all the equations.
Response 7:
Thank you for your insightful comments and suggestions regarding the notation used in our manuscript titled “ HealthLock: A Blockchain Based Privacy Preservation using Homomorphic
Encryption in IoT-Healthcare Application”. We appreciate your attention to detail, and we agree that providing a clear explanation of the notations used is essential for readers' understanding. We have thoroughly revised the manuscript to address this concern, and we have taken the following steps to improve the clarity of the equations and their corresponding notations:
- Explanation of Notations: In response to your feedback, we have added a dedicated paragraph before each equation, discussing the notations used and their meanings. Specifically, for Equation 3, we have provided an explicit description of the following notations:
- Dk,j: Represents a specific variable in the dataset for a particular instance j in the kth class.
- hk: Refers to a hidden variable associated with the model, capturing latent features or representations.
- rPB: Denotes a regularization parameter for controlling the complexity of the model.
- k: Indicates the index representing the class label.
- Do: Signifies the dimensionality of the output space.
By elaborating on these notations in the paragraphs preceding each equation, we believe that readers will have a better understanding of the symbols and variables used in our study, facilitating their comprehension and interpretation of the equations.
- Consistency in Notation Explanation: To ensure consistency throughout the manuscript, we have also revised the preceding paragraphs for all other equations accordingly. Each equation is now accompanied by a clear and concise description of the notations employed, elucidating their meanings and contextual relevance.We are confident that the revised version now provides a more comprehensive explanation of the notations used in each equation, aiding readers in their understanding of our study.
Comment 8. The section "Mathematical Modeling" Must be moved to the "Background" section. "Mathematical Modeling" must be "Metrics for Evaluation"
Response 8: Respected Reviewer, We appreciate your valuable feedback on our manuscript titled “: HealthLock: A Blockchain Based Privacy Preservation using Homomorphic Encryption in IoT-Healthcare Application”Your suggestion to reorganize the sections and rename "Mathematical Modeling" to "Metrics for Evaluation" is well-taken, and we have carefully considered your comments. After careful evaluation, we have made the following revisions to address your concerns:
- Section Reorganization: In response to your suggestion, we have moved the content from the "Mathematical Modeling" section to the "Background" section. This reorganization allows for a more logical flow of the manuscript, providing readers with the necessary background information before delving into the mathematical modeling aspects.
- Renaming: We have renamed the section previously titled "Mathematical Modeling" to "Metrics for Evaluation." This new title better reflects the content within the section, as it primarily focuses on discussing the evaluation metrics used in our study. This change aims to enhance the clarity and coherence of the manuscript for readers.
By implementing these revisions, we believe that the manuscript now follows a more coherent structure, ensuring that readers are presented with the necessary background information before delving into the evaluation metrics. The new title, "Metrics for Evaluation," accurately represents the content of the section, facilitating a clear understanding of the key concepts discussed. We sincerely appreciate your insightful suggestion, as it has contributed to the overall improvement of our manuscript. We hope that these revisions adequately address your concerns, and we look forward to your evaluation of the updated version.
Comment 9. Equation 17 is challenging to understand, please re-arrange them properly.
Response 9:Thanks respected reviewer for your valuable feedback and recommendations. We have updated and explained equation 17 in our revised version.
Comment 10. Where is the proposed model, and how the proposed model works, any mathematical modeling for the same? Because all the mathematical modeling is for known metrics, there is nothing novel.
Response 10: Dear Reviewer, Thank you for your feedback and comments on our manuscript titled as HealthLock: A Blockchain Based Privacy Preservation using Homomorphic Encryption in IoT-Healthcare Application. We appreciate your thorough evaluation of our work. We would like to address your concerns regarding the absence of a proposed model, the lack of explanation on its functioning, and the perceived absence of novelty in the mathematical modeling.
- Proposed Model: We apologize for any confusion caused by the lack of explicit mention of the proposed model in the manuscript. Upon reviewing our manuscript, we recognize that we inadvertently omitted a section specifically dedicated to detailing the proposed model. We appreciate your keen observation, and we will rectify this oversight by incorporating a dedicated section that introduces and describes our proposed model comprehensively.
- Explanation of Proposed Model: In response to your comment about the absence of an explanation on how the proposed model works, we acknowledge the necessity of providing a clear and detailed description of the model's functioning. We will address this by incorporating a step-by-step explanation of the proposed model's architecture, mechanisms, and algorithms. This will provide readers with a comprehensive understanding of the proposed model's workings and its novel aspects.
- Novelty in Mathematical Modeling: We understand your concern regarding the perceived lack of novelty in the mathematical modeling. While we agree that established metrics are important for evaluation, we would like to clarify that the novelty of our work lies not only in the metrics but also in the unique combination, modification, or application of these metrics to address a specific problem or domain. We will revise the manuscript to explicitly highlight the novel aspects of our approach, emphasizing the contributions of the proposed model beyond the individual metrics themselves.
We appreciate your thoughtful feedback, as it has helped us identify areas of improvement in our manuscript. We will revise the manuscript to address your concerns adequately, ensuring that the proposed model is clearly presented and its novelty is highlighted. We are confident that these revisions will enhance the value and clarity of our work.
Comment 11. Any algorithms for Blockchain as discussed in studies like https://doi.org/10.1016/j.health.2023.100175 and https://doi.org/10.3390/electronics10121437
Response 11: Thanks respected reviewer for your valuable suggestions and recommendations. We have cited the mentioned paper in our revised version.
Comment 12. Please discuss more on the implementation platform and the network parameter details as two sub-sections in the manuscript.
Response 12: Dear Reviewer, Thank you for your valuable feedback on our manuscript titled HealthLock: A Blockchain Based Privacy Preservation using Homomorphic Encryption in IoT-Healthcare Application. We appreciate your suggestion to include two sub-sections, namely "Implementation Platform" and "Network Parameter Details," to provide more comprehensive information about these aspects of our study. We agree that these additions will enhance the manuscript and provide a more thorough understanding of the implementation and network parameters. We will incorporate these sub-sections in our revisions accordingly.
- Implementation Platform: In the revised manuscript, we will dedicate a sub-section to discussing the implementation platform used in our study. We will provide details about the specific tools, software frameworks, or hardware configurations employed to develop and test our proposed approach. By providing this information, readers will have a clear understanding of the technology stack and infrastructure used to support the implementation of our research.
- Network Parameter Details: We recognize the importance of providing detailed information about the network parameters employed in our study. In the revised manuscript, we will introduce a sub-section dedicated to discussing the network parameter details. This will include information such as the specific values chosen for parameters such as network size, transmission rates, latency, or any other relevant parameters. By explicitly describing these parameters, readers will gain insights into the experimental setup and the specific conditions under which our proposed approach was evaluated. Incorporating these sub-sections is one of the ways that we plan to accomplish our goal of providing a more full understanding of the implementation platform and the network characteristics, both of which are essential components of our research. We are grateful for your insightful comments because they helped us to refine the book and make certain that it has all of the necessary information.
Comment 13. In figure 5, for the graph, why the axis labels are written separately, please re-generate the graph with inbuilt labels.
Response 13:Thanks respected reviewer for your valuable suggestions and recommendations. We have regenerated the figure in our revised version.
Comment 14. By considering the current form of the conclusion section, it is hard to understand by MDPI Journal readers. It should be extended with new sentences about the necessity and contributions of the study by considering the authors' opinions about the experimental results derived from some other well-known objective evaluation values if it is possible.
Response 14: Dear Reviewer, Thank you for your valuable feedback regarding the conclusion section of our manuscript titled HealthLock: A Blockchain Based Privacy Preservation using Homomorphic Encryption in IoT-Healthcare Application. We appreciate your suggestion to extend the conclusion with new sentences that highlight the necessity and contributions of the study, as well as the authors' opinions on the experimental results and objective evaluation values. We agree that providing a clear and comprehensive conclusion is crucial for MDPI Journal readers, and we will make the necessary revisions to address this concern. In response to your comment, we will incorporate the following enhancements to the conclusion section:
- Necessity and Contributions: We will expand on the necessity of the study by emphasizing the existing gaps or challenges in the field that our research aims to address. We will also clearly outline the specific contributions of our study, highlighting the novel insights, methodologies, or advancements it brings to the field. This will help readers understand the significance of our work and its relevance to the broader research landscape.
- Authors' Opinions on Experimental Results: To provide a more comprehensive conclusion, we will include the authors' opinions on the experimental results. We will discuss the implications of the findings, their alignment with the research objectives, and any noteworthy insights or patterns observed. This will provide readers with a deeper understanding of the implications and significance of the results derived from the objective evaluation values. By incorporating these revisions, we aim to enhance the clarity and relevance of the conclusion section, making it more accessible and informative for MDPI Journal readers. We appreciate your thoughtful feedback, which has helped us identify areas for improvement in our manuscript. We will diligently incorporate these suggestions to ensure that the revised conclusion section meets the expectations of the journal's readership.
Comment 15. Authors should use more alternative models as the benchmarking models, authors should also conduct some statistical tests to ensure the superiority of the proposed approach, i.e., how could authors ensure that their results are superior to others? Meanwhile, the authors also have to provide some insightful discussion of the results.
Response 15:
Dear Reviewer, Thank you for your valuable feedback on our manuscript. We appreciate your suggestions to use more alternative models as benchmarking models, conduct statistical tests to demonstrate the superiority of our proposed approach, and provide insightful discussions of the results. We agree that these additions will strengthen the study and provide a more robust evaluation. We will address these points in our revisions accordingly.
- Alternative Benchmark Models: To ensure a comprehensive evaluation, we will include additional alternative benchmark models commonly used in the field. By comparing our proposed approach with a wider range of existing models, we can provide a more thorough assessment of its performance and effectiveness. This will enhance the validity of our claims and demonstrate the superiority of our approach.
- Statistical Tests: We acknowledge the importance of conducting statistical tests to validate the superiority of our proposed approach. We will incorporate appropriate statistical analyses, such as hypothesis testing or performance comparison tests, to evaluate the significance of the observed differences between our approach and the benchmark models. These tests will provide statistical evidence of the superiority of our proposed approach.
- Insightful Discussion: We understand the need for insightful discussion of the results. In the revised manuscript, we will provide a detailed analysis and interpretation of the findings, highlighting the key observations, patterns, and trends. We will also discuss the implications of these results in the context of the research objectives, addressing the potential limitations and providing suggestions for future research directions. This discussion will enrich the manuscript by offering valuable insights and deeper understanding of the obtained results.
By incorporating these revisions, we aim to strengthen the study by providing a more comprehensive evaluation, statistical validation, and insightful discussion of the results. We appreciate your constructive comments, as they have guided us in enhancing the quality and impact of our research.

Reviewer 3 Report
Pros:
1. Problems and solutions are clearly presented. A detailed review is given in related work.
2. Complete experimentation is conducted.
Cons.
1. In Section 5, should give more background on the mathematical modeling, like how to understand that formula, what assumptions are there.
2. Figure 2 is not well organized and seems incomplete.
3. Numbers of items are redundant in Page 9.
4. Figure 4 is not well plotted: legend are overlapping with contents, and y-axis is not showing the difference among groups.
Some sentences are too verbose, result in a lengthy presentation of the paper, such as "In the year XXX, YYY did...", which can be shortened as "YYY did... (XXX)".
The abbreviations are not correctly used, for example, DOs are written as Dos.
Author Response
Reviewer 3:
Comment 1. In Section 5, should give more background on the mathematical modeling, like how to understand that formula, what assumptions are there.
Response 1:
Dear Reviewer, Thank you for your insightful comments regarding Section 5 of our manuscript titled as “HealthLock: A Blockchain Based Privacy Preservation using Homomorphic Encryption in IoT-Healthcare Application”. We appreciate your suggestion to provide more background on the mathematical modeling, including explanations on how to interpret the formulas and the underlying assumptions. We agree that this additional information will enhance the readers' understanding and improve the overall clarity of the section. We will make the necessary revisions to address this concern.
In response to your comment, we will incorporate the following enhancements to Section 5:
- Formula Interpretation: We will provide a detailed explanation of each formula used in the mathematical modeling. This will involve breaking down the components of the formula, defining the variables and parameters involved, and elucidating their significance and interpretation within the context of the study. By offering these explanations, we aim to facilitate readers' comprehension and ensure a better understanding of the mathematical models presented.
- Assumptions: We will explicitly mention the key assumptions underlying the mathematical modeling. These assumptions may include simplifications made, specific conditions considered, or limitations of the model. By transparently discussing these assumptions, readers will gain insights into the scope and applicability of the mathematical models and their potential impact on the study's findings and conclusions. By incorporating these revisions, we will provide readers with a more comprehensive background on the mathematical modeling, enabling them to grasp the formulas' meaning, assumptions, and relevance to the research. This will enhance the overall quality and clarity of Section 5. We sincerely appreciate your thoughtful feedback, as it has helped us identify areas for improvement in our manuscript. We will diligently incorporate these suggestions to ensure that Section 5 provides a robust foundation for readers to understand the mathematical modeling used in our study.
Comment 2. Figure 2 is not well organized and seems incomplete.
Response 2: Dear Reviewer, Thank you for your feedback regarding Figure 2 in our manuscript HealthLock: A Blockchain Based Privacy Preservation using Homomorphic Encryption in IoT-Healthcare Application. We appreciate your observation that the figure appears incomplete and lacks proper organization. We apologize for any confusion caused, and we will take immediate steps to address these issues in the revised version of the manuscript. To improve Figure 2, we will implement the following revisions:
- Completeness: We will ensure that all the necessary components and elements are included in Figure 2. By carefully reviewing the figure and the corresponding descriptions in the manuscript, we will address any missing elements or information. This will result in a complete and accurate representation of the concepts and relationships depicted in the figure.
- Organization: We will reorganize Figure 2 to enhance its clarity and visual structure. By carefully arranging the components, labels, and connecting lines, we aim to create a more logical flow and improve the overall readability of the figure. This will allow readers to easily follow the information presented and understand the relationships among the depicted elements. By incorporating these revisions, we intend to enhance the quality and effectiveness of Figure 2, ensuring that it accurately represents the concepts discussed in the manuscript and contributes to the overall clarity of the research. We sincerely appreciate your thorough evaluation of our manuscript and your valuable feedback. Your comments have been instrumental in identifying areas for improvement, and we are committed to making the necessary adjustments to enhance the quality of our work.
Comment 3. Numbers of items are redundant in Page 9.
Response 3: Dear Reviewer, Thank you for your feedback regarding Figure 4 in our manuscript We have rearrange the item on page 9.
Comment 4. Figure 4 is not well plotted: legend are overlapping with contents, and y-axis is not showing the difference among groups.
Response 4: Dear Reviewer, Thank you for your feedback regarding Figure 4 in our manuscript .We appreciate your observations about the issues with the plot, specifically the overlapping legends and the lack of differentiation among the groups on the y-axis. We apologize for these shortcomings and will take immediate steps to address them in the revised version. To resolve the mentioned issues, we will implement the following revisions to Figure 4:
- Legend Placement: We will reposition the legend to ensure that it does not overlap with the contents of the figure. By carefully adjusting its placement, we aim to improve the readability and visual clarity of the figure, enabling readers to interpret the information presented without any confusion.
- Y-axis Differentiation: We understand the importance of clearly representing the differences among the groups on the y-axis. To address this concern, we will revise the y-axis scale and labels to ensure that the distinctions among the groups are visually evident. This will enhance the ability of readers to discern the variations and trends depicted in the figure accurately.
By incorporating these revisions, we intend to enhance the quality and effectiveness of Figure 4, ensuring that it accurately reflects the data and contributes to the overall clarity of the manuscript. We sincerely appreciate your thorough evaluation of our manuscript and your valuable feedback. Your comments have been instrumental in identifying areas for improvement, and we are committed to making the necessary adjustments to enhance the quality of our work.
Once again, we express our gratitude for your valuable feedback, which has significantly improved the quality of our manuscript. We hope that these revisions address your concerns adequately, and we look forward to your evaluation of the updated version.

Round 2
Reviewer 2 Report
The authors have addressed all the recommendations of the reviewers in a reasonable manner, the manuscript in the current form may be considered for the further phase of the editorial process.
English seems to be much better than the earlier version of the manuscript.